# Skewness-Robust Causal Discovery in Location-Scale Noise Models

**Daniel Klippert** [1] [2]   **Alexander Marx** [1] [2]

## Abstract

To distinguish Markov equivalent graphs in causal discovery, it is necessary to restrict the structural causal model. A flexible class of models that is general and identifiable in most cases are location-scale noise models (LSNMs), in which the effect $Y$ is modeled based on its causes $\boldsymbol{X}$ as $Y = f(\boldsymbol{X}) + g(\boldsymbol{X})N$. To facilitate the estimation of these models, a prominent assumption is that the noise variable $N$ follows a symmetric distribution. We show that when $N$ is a skewed random variable, which is likely in real-world domains, such approaches drop in performance. To address this limitation, we propose SKEWD, a likelihood-based method for causal discovery under LSNMs with skewed noise, employing a combination of heuristic search and expectation conditional maximization for parameter estimation. SKEWD extends the usual normal distribution framework to the skew-normal setting, enabling reliable inference under symmetric and skewed noise. While our main focus is on bivariate cause-effect inference, we further showcase how SKEWD can be extended to the multivariate setting.

## 1. Introduction

Causal discovery describes the process of learning cause-effect relationships from observational data and is of strong interest across most scientific fields, including Earth sciences (Runge et al., 2019), neuroscience (Ramsey et al., 2010) and manufacturing (Vuković & Thalmann, 2022). We study causal discovery from independently and identically distributed (iid) data and under causal sufficiency, concentrating on approaches that restrict the structural causal model (SCM). Compared to classical constraint-based or score-

[1]Department of Statistics, TU Dortmund University, Germany [2]Research Center Trustworthy Data Science and Security, University Alliance Ruhr, Germany. Correspondence to: Daniel Klippert <daniel.klippert@tu-dortmund.de>, Alexander Marx <alexander.marx@tu-dortmund.de>.

*Proceedings of the $43^{rd}$ International Conference on Machine Learning*, Seoul, South Korea. PMLR 306, 2026. Copyright 2026 by the author(s).

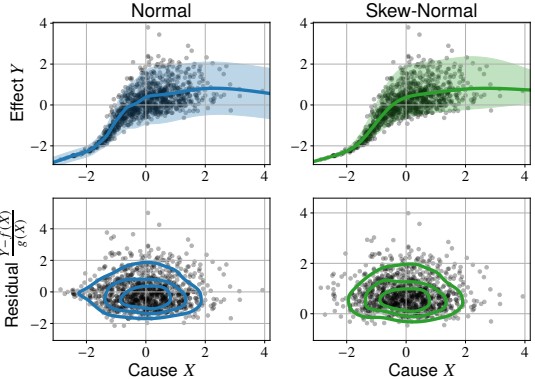

*Figure 1.* Comparison of LSNM mean fits and confidence intervals based on the normal distribution via LOCI (left) and skew-normal distribution via SKEWD (right) for Pair 6 of the new LSs(1.750) dataset. The estimated residuals in the normal-model display a dependence with cause $X$, which stems from an inadequate variance fit for small values of $x$. The dependence is indicated by the pointy outer contour level of the kernel density estimate, leading to a wrong inference. SKEWD overcomes this limitation and infers the correct direction.

based methods (Spirtes et al., 2000; Chickering, 2002), such restrictions enable identifiability beyond Markov equivalence classes. In this paper, we consider the broad class of location-scale noise models (LSNMs), where the relationship between causes $\boldsymbol{X}$ and effect $Y$ is described by $Y = f(\boldsymbol{X}) + g(\boldsymbol{X})N$, with strictly positive $g$ and noise $N$ that is independent of $\boldsymbol{X}$. LSNMs generalize additive noise models (ANMs) and have been shown to be identifiable except for pathological cases (Strobl & Lasko, 2023; Immer et al., 2023). We primarily focus on the bivariate setting with data from two random variables $X$ and $Y$, where causal discovery reduces to deciding whether $X$ causes $Y$ or $Y$ causes $X$. This setting allows for a controlled analysis of model misspecification, while serving as a foundation for multivariate extensions (Peters et al., 2014).

Despite the generality of the identification result for LSNMs, prior work predominantly modeled symmetric noise distributions in bivariate (Immer et al., 2023; Strobl & Lasko, 2023; Tran et al., 2024) and multivariate models (Duong & Nguyen, 2023; Lin et al., 2024). This assumption is, however, often violated in real-world data, e.g., Genton (2004) provides numerous data examples exhibiting skewed

distributions, including coastal flooding, spatial rainfall prediction and stochastic frontier analysis. Skewness induces a misspecification in likelihood models that assume symmetric noise, which in turn complicates estimation of $f$ and $g$. In Figure 1, we illustrate the problem that skewed noise distributions pose to these approaches. Here, we apply LOCI (Immer et al., 2023), which assumes Gaussian noise terms and follows a RESIT-type approach (Peters et al., 2014) to a skewed LSNM. Such approaches, aim to extract the noise term and subsequently test for independence with the hypothetical cause. Even small distortions in the estimated functions can propagate to the residuals hindering independence between the cause and residual, which would confirm the model assumptions. Here, we see that the misspecification leads to a dependence, and consequently, independence is falsely rejected.

As demonstrated in our experiments, this misspecification issue persists when resorting to likelihood-based approaches. This observation is in line with prior work that highlighted the pitfalls of Gaussian likelihood scoring in causal discovery for linear ANMs (Reisach et al., 2021) and LSNMs (Schultheiss & Bühlmann, 2023). To address this limitation, we propose SKEWD, a causal discovery approach for LSNMs that models the noise via the skew-normal distribution (Azzalini, 1985), a generalization of the Gaussian. In the bivariate example provided in Figure 1, we can see that SKEWD effectively recovers the independent noise variable, thus inferring the correct causal direction. Concretely, our main contributions are as follows:

- We introduce the skew-normal LSNM (Section 3) and derive an estimation procedure based on an expectation conditional maximization algorithm (Section 4).
- We propose SKEWD as a practical approach to cause-effect inference (bivariate) via residual independence testing or likelihood scoring, and extend the likelihood scoring variant to the multivariate setting (Section 5).
- We empirically compare SKEWD to state-of-the-art (SOTA) algorithms on established and novel bivariate benchmarks with varying skewness levels, as the bivariate setting allows for a controlled assessment of skewness effects, and additionally evaluate the multivariate extension.

We find that SKEWD stands out as the most reliable approach for LSNMs under high skewness while performing on par with SOTA in other settings.[1]

## 2. Related Work

**Causal Discovery** We here focus on methods that restrict the structural causal model (SCM) to distinguish Markov

---

[1]To ensure reproducibility, the implementation of our method and experimental runs are publicly available under `https://github.com/causality-lab/skewd`.

equivalent DAGs, where the central problem is to solve the bivariate problem. Identification results for a variety of function classes besides LSNMs have been derived, including additive noise models (ANMs) (Shimizu et al., 2006; Hoyer et al., 2008; Bühlmann et al., 2014) and post-nonlinear additive noise models (PNL) (Zhang & Hyvärinen, 2009; Zhang et al., 2015; Uemura et al., 2022), where PNL and LSNMs are both generalizations of ANMs. Besides those approaches, some works focus on low-noise distributions (almost deterministic) in the context of ANMs (Janzing et al., 2012; Blöbaum et al., 2018), and LSNMs (Cai et al., 2020; Xu et al., 2022). Another body of work aims to exploit the postulate of independent mechanisms by either instantiating it directly (Sgouritsa et al., 2015; Dhir et al., 2024), or by approximating the algorithmic Markov condition (Janzing & Schölkopf, 2010) by compression-based methods (Mooij et al., 2010; Marx & Vreeken, 2017; Tagasovska et al., 2020; Mian et al., 2021; Marx & Vreeken, 2022).

LSNMs are a rather broad model class, for which general identification results exist (Immer et al., 2023; Strobl & Lasko, 2023), implying that LSNMs are identifiable in most cases. Although an exhaustive enumeration of non-identifiable cases does not exist, for Gaussian noise (Khemakhem et al., 2021), exponential family models (Bodik & Chavez-Demoulin, 2025), uniform and continuous Bernoulli (Sun & Schulte, 2023), the non-identifiable cases have been characterized. Another approach, which builds upon the identification result of LOCI (Immer et al., 2023), relaxes the Gaussianity assumption for the parameter estimation is ROCHE (Tran et al., 2024), where the authors model the noise distribution via Student's $t$-distribution. Some of the approaches above have been extended to the *multivariate setting* (Duong & Nguyen, 2023; Kamkari et al., 2024). Further, DAG learning has been accelerated by introducing differentiable DAG constraints (Zheng et al., 2018). From this class of models, GraN-DAG (Lachapelle et al., 2020) can be extended to LSNMs, whereas most methods focus on ANMs (Vowels et al., 2022). Notably, Lin et al. (2024) show how under the assumption of symmetric noise, skewness will appear in the anti-causal model, which allows them to derive a method based on score-matching (Rolland et al., 2022). In contrast, we explicitly model the noise as a skewed random variable and develop both an independence-based and a likelihood-based approach.

**Skew-Normal** The skew-normal distribution (Azzalini, 1985) is a popular choice for modeling skewness, though maximum likelihood estimation is challenging. As noted by Azzalini & Arellano-Valle (2013), its likelihood may lack a finite maximum, which can be addressed by including a penalty term with specific properties. Another difficulty is a stationary point at $\lambda = 0$ when considering the parameterization as defined in Equation (2). To circumvent this problem, maximization can be carried out with respect to a reparame-

terization of the skew-normal (Azzalini, 1985). Moreover, expectation maximization (EM) algorithms (Azzalini, 2013) and genetic algorithms have been considered (Yalçınkaya et al., 2018). Ferreira & Paula (2017) developed an EM algorithm for partially linear models that was extended to an expectation conditional maximization (ECM) algorithm for heteroscedastic partially linear models by Ferreira et al. (2023). We adapt their approach to LSNMs, extend function classes to generalized additive models and use it to improve the result of a heuristic optimization.

## 3. Modeling Skewness in LSNMs

In this section, we formally introduce the location-scale noise model and define the notion of skewness. We then characterize the skew-normal distribution and specify the skew-normal LSNM used in our framework. Finally, we motivate the use of the skew-normal distribution over alternative assumptions.

### 3.1. Skew-Normal LSNMs

**Definition 3.1** (**Location-Scale Noise Models**). Consider a structural causal model (SCM) over $d$ continuous random variables $Y^{(1)}, \ldots, Y^{(d)}$ with associated directed acyclic graph (DAG) $\mathcal{G}$ and assume structural minimality. The SCM is called a location-scale noise model (LSNM) if for each variable $Y^{(j)}$ with non-empty parent set $\mathrm{PA}_{Y^{(j)}}^{\mathcal{G}} \neq \emptyset$, the structural equation of $Y^{(j)}$ is given by

$$Y^{(j)} = f^{(j)}(\mathrm{PA}_{Y^{(j)}}^{\mathcal{G}}) + g^{(j)}(\mathrm{PA}_{Y^{(j)}}^{\mathcal{G}})N_{Y^{(j)}}, \quad (1)$$

where $f^{(j)}$ and $g^{(j)}$ are deterministic real-valued functions with $g^{(j)}(\cdot) > 0$ and $N_{Y^{(j)}}$ is a noise variable. If $\mathrm{PA}_{Y^{(j)}}^{\mathcal{G}} = \emptyset$, then $Y^{(j)} = N_{Y^{(j)}}$. The noise variables $N_{Y^{(1)}}, \ldots, N_{Y^{(d)}}$ are jointly independent.

**Definition 3.2** (**Skewness**). The skewness of a univariate random variable $X$ with mean $\mathbb{E}[X] = \mu$ and variance $\mathbb{E}[(X - \mu)^2] = \sigma^2$ corresponds to the third centralized moment and is given by

$$\gamma_1 = \mathbb{E}[(X - \mu)^3/\sigma^3].$$

$X$ is said to be skewed or asymmetric if $\gamma_1 \neq 0$. If $\gamma_1 > 0$, $X$ is right-skewed, and if $\gamma_1 < 0$, it is left-skewed.

**Definition 3.3** (**Skew-Normal Distribution**). The random variable $X$ follows a univariate skew-normal distribution (Azzalini, 1985) with parameters $\xi \in \mathbb{R}$ (location), $\omega > 0$ (scale), and $\lambda \in \mathbb{R}$ (shape), $X \sim \mathrm{SN}(\xi, \omega, \lambda)$, if its probability density function (pdf) is given by

$$p(x) = \frac{2}{\omega} \phi\left(\frac{x - \xi}{\omega}\right) \Phi\left(\lambda \frac{x - \xi}{\omega}\right), \quad x \in \mathbb{R}, \quad (2)$$

where $\phi$ and $\Phi$ correspond to the pdf and cumulative distribution function (cdf) of the standard normal distribution.

The $\mathrm{SN}(\xi, \omega, \lambda)$ generalizes the normal distribution, since it reduces to a regular normal distribution with mean $\xi$ and variance $\omega^2$ if $\lambda = 0$. It is right-skewed if $\lambda > 0$ and left-skewed if $\lambda < 0$. Its skewness is approximately bounded by the interval $(-\gamma_1^{\max}, \gamma_1^{\max})$, where $\gamma_1^{\max} \approx 0.9953$ (Azzalini, 2013). We provide additional details in Appendix A.

**Skew-Normal LSNM** The assumption of symmetric error distributions in existing cause-effect inference methods for LSNMs (Immer et al., 2023; Tran et al., 2024; Lin et al., 2024) may limit their practical applicability. To ensure reliable cause-effect and causal graph learning in the presence of skewed noise, we propose to model the noise in Equation (1) via the skew-normal distribution $N_{Y^{(j)}} \sim \mathrm{SN}(0, 1, \lambda)$. Under this assumption, our goal is to infer the causal graph from $n$ independent and identically distributed (iid) samples of the random vector $(Y^{(1)}, \ldots, Y^{(d)})^\top$.

For simplicity, we drop the superscript $j$ and define the cause-effect model for a fixed variable $Y := Y^{(j)}$ with corresponding noise $N_Y := N_{Y^{(j)}}$. Let $\boldsymbol{X} = (X^{(1)}, \ldots, X^{(s)})^\top$ with $s > 0$ denote the vector of the variables in the parent set $\mathrm{PA}_{Y^{(j)}}^{\mathcal{G}}$ of the considered graph $\mathcal{G}$. By the density transformation formula it follows that the conditional distribution of the effect given its cause(s) is skew-normally distributed with $Y|\boldsymbol{X} = \boldsymbol{x} \sim \mathrm{SN}(f(\boldsymbol{x}), g(\boldsymbol{x}), \lambda)$. To allow for flexible functional forms, we model both $f$ and $g$ using generalized additive models (GAMs) (Hastie & Tibshirani, 1986), where each covariate is represented by an individual B-spline basis (Eilers & Marx, 1996).

The resulting compact cause-effect model for $Y_i$ given $\boldsymbol{x}_i$, $i = 1, \ldots, n$, is

$$Y_i = f(\boldsymbol{x}_i) + g(\boldsymbol{x}_i)N_{Y_i} = \mathbf{n}_i^\top \boldsymbol{\psi} + \exp\left(0.5\mathbf{z}_i^\top \boldsymbol{\rho}\right) N_{Y_i}, \quad (3)$$

where $\mathbf{n}_i^\top \in \mathbb{R}^{1 \times q}$, $\mathbf{z}_i^\top \in \mathbb{R}^{1 \times p}$ correspond to the $i$-th row of the design matrices $\mathbf{N} \in \mathbb{R}^{n \times q}$ and $\mathbf{Z} \in \mathbb{R}^{n \times p}$, which combine the covariate-specific spline bases. The corresponding unknown coefficient vectors are denoted by $\boldsymbol{\psi} \in \mathbb{R}^q$, and $\boldsymbol{\rho} \in \mathbb{R}^p$. Further details on the GAM construction and the associated matrices are provided in Appendix A. Similar to the approach of Ferreira et al. (2023), we apply the exponential function to ensure positivity of the estimated scale parameter. Inferring the causal graph requires estimating the above model for multiple variables $Y^{(j)}$ (see Section 5). We discuss parameter estimation in Section 4.

### 3.2. On the Choice of Noise Distribution

We chose to model the noise using the skew-normal due to several desirable properties. First, unlike distributions such as the exponential, log-normal, gamma, or generalized normal (GNO) (Hosking, 1986), the skew-normal has unbounded support over the real line, making it suitable for modeling LSNMs. Second, it generalizes the Gaus-

sian, which is a standard assumption for noise modeling in many applications, making it a more natural choice than alternative unbounded distributions such as the skew-$t$ distribution (Azzalini & Arellano-Valle, 2013). While the skew-$t$ distribution can model higher levels of skewness, it also introduces an additional degrees of freedom parameter and its skewness coefficient is only defined for degrees of freedom greater than three. Third, the skew-normal provides sufficient flexibility to capture relevant asymmetries, while maintaining a relatively simple parametric form. As demonstrated in Section 6, its practical applicability extends even to settings that exceed its theoretical skewness and excess kurtosis range. While skew-elliptical densities or nonparametric approaches offer greater modeling capabilities, they may be too flexible to guarantee accurate estimation. As demonstrated in the experiments, nonparametric approaches lead to unsatisfactory performance in skewed LSNMs.

## 4. Parameter Estimation

To estimate the model for $Y|\boldsymbol{X} = \boldsymbol{x}$ within the skew-normal LSNM, we employ a maximum likelihood approach. In the following, we define the (penalized) log-likelihood and discuss our estimation approach, which includes an expectation conditional maximization algorithm.

**Model Log-Likelihood** From Equation (3) it follows that the observed log-likelihood of the effect $\boldsymbol{y} = (y_1, \ldots, y_n)^\top$ given the cause(s) $\boldsymbol{x}_i$ with respect to the parameters $\boldsymbol{\theta} = (\boldsymbol{\psi}^\top, \boldsymbol{\rho}^\top, \lambda)^\top$ is given by

$$\ell(\boldsymbol{\theta}) = \frac{n}{2} \log\left(\frac{2}{\pi}\right) - \frac{1}{2} \sum_{i=1}^{n} \mathbf{z}_i^\top \boldsymbol{\rho} - \frac{1}{2} \sum_{i=1}^{n} \frac{(y_i - \mathbf{n}_i^\top \boldsymbol{\psi})^2}{\exp(\mathbf{z}_i^\top \boldsymbol{\rho})}$$
$$+ \sum_{i=1}^{n} \log\left(\Phi\left(\lambda \frac{y_i - \mathbf{n}_i^\top \boldsymbol{\psi}}{\exp(0.5 \mathbf{z}_i^\top \boldsymbol{\rho})}\right)\right).$$

**Penalized Log-Likelihood** To prevent an overfitting behavior, we adopt a penalized log-likelihood approach that imposes roughness penalties based on second-order differences of the spline coefficients (Eilers & Marx, 1996). For covariates $t = 1, \ldots, s$, the coefficients $\boldsymbol{\psi}$ and $\boldsymbol{\rho}$ are penalized through covariate-specific parameters $\boldsymbol{\alpha} = (\alpha_1, \ldots, \alpha_s)^\top$ and $\boldsymbol{\kappa} = (\kappa_1, \ldots, \kappa_s)^\top$, respectively, with $\alpha_t, \kappa_t > 0$. The penalties are expressed via matrices $\mathbf{K}_{\boldsymbol{\alpha}} \in \mathbb{R}^{q \times q}$ and $\mathbf{M}_{\boldsymbol{\kappa}} \in \mathbb{R}^{p \times p}$ given $\boldsymbol{\alpha}$ and $\boldsymbol{\kappa}$. The resulting penalized log-likelihood takes the form

$$\ell_p(\boldsymbol{\theta}, \boldsymbol{\alpha}, \boldsymbol{\kappa}) = \ell(\boldsymbol{\theta}) - \frac{1}{2} \boldsymbol{\psi}^\top \mathbf{K}_{\boldsymbol{\alpha}} \boldsymbol{\psi} - \frac{1}{2} \boldsymbol{\rho}^\top \mathbf{M}_{\boldsymbol{\kappa}} \boldsymbol{\rho}. \quad (4)$$

We state the explicit expressions for the second-order difference penalties on $\boldsymbol{\psi}$ and $\boldsymbol{\rho}$ given $\boldsymbol{\alpha}$ and $\boldsymbol{\kappa}$ in Appendix A. In our causal discovery method, we estimate the model parameters of the LSNM in Equation (3) by maximizing the penalized log-likelihood from Equation (4).

---

**Algorithm 1** Estimate model parameters $\boldsymbol{\theta}$

1: Perform Bayesian optimization to find optimal $\widehat{\boldsymbol{\alpha}}, \widehat{\boldsymbol{\kappa}}$
2: Find $\widehat{\boldsymbol{\theta}}_{\mathrm{h}} = \mathrm{argmax}_{\boldsymbol{\theta}} \, \ell_p(\boldsymbol{\theta}, \widehat{\boldsymbol{\alpha}}, \widehat{\boldsymbol{\kappa}})$ via CMA-ES
3: Submit $\widehat{\boldsymbol{\theta}}_{\mathrm{h}}$ as initial value to ECM algorithm yielding final parameter estimate $\widehat{\boldsymbol{\theta}}$

---

**Challenges in Optimization** Maximizing the penalized log-likelihood defined in Equation (4) involves two challenges. First, $\boldsymbol{\alpha}$ and $\boldsymbol{\kappa}$ must be set adequately to balance model fit and regularization. Secondly, given fixed $\boldsymbol{\alpha}$ and $\boldsymbol{\kappa}$, a reliable maximization approach for this non-convex optimization problem is required, which is difficult for the skew-normal model, as discussed in Section 2. We base our optimization on the ECM algorithm by Ferreira et al. (2023). An advantage of EM type algorithms is that they increase or maintain the likelihood with every update (Dempster et al., 1977; Meng & Rubin, 1993), which carries over when the penalized likelihood is maximized (Green, 1990). However, slow convergence and sensitivity to local optima often necessitate multiple initializations (Karlis & Xekalaki, 2003), which is computationally costly in our $(p + q + 1)$-dimensional parameter space.

**Optimization Approach** The main steps of our estimation approach are summarized in Algorithm 1. Instead of directly using an ECM algorithm, assuming for now we have fixed values for $\boldsymbol{\alpha}$ and $\boldsymbol{\kappa}$, we heuristically optimize the penalized log-likelihood using the covariance matrix adaptation evolution strategy (CMA-ES). The CMA-ES is an evolutionary optimization algorithm for non-linear, non-convex functions (Hansen & Ostermeier, 2001; Hansen, 2016). Since it is stochastic, we use multiple starting values for $\boldsymbol{\theta}$ and submit the best result to an ECM algorithm to refine the obtained heuristic solution (lines 2–3). In Appendix B, we show how the ECM algorithm improves upon the performance achieved by the initial heuristic solution in the bivariate causal discovery task on our synthetic data, demonstrating the necessity of refinement. Since the described approach requires fixed penalty parameters, we precede it with a Bayesian optimization step (line 1) to find adequate values for $\boldsymbol{\alpha}$ and $\boldsymbol{\kappa}$. We refer to Appendix B for more details on implementation and hyperparameter settings used in our experiments (Section 6).

**Expectation Conditional Maximization (ECM)** ECM algorithms (Meng & Rubin, 1993) iteratively derive maximum likelihood estimates within latent variable models by alternating between an expectation step (E-step) and a sequence of conditional maximization steps (CM-steps). In each iteration, the likelihood is increased (or maintained) by optimizing a lower bound, known as the $Q$-function which is computed in the E-step. It is the expected value of the complete data log-likelihood over the conditional distribution

of the latent variables given the observed data and current parameter estimates. In the CM-steps, the parameter vector $\boldsymbol{\theta}$ is updated by sequentially maximizing the $Q$-function with respect to subsets of the parameters while keeping the remaining parameter fixed.

In the following, we outline the ECM algorithm employed to refine the heuristic parameter estimate for $\boldsymbol{\theta}$. It is based on the ECM algorithm for partially nonlinear models by Ferreira et al. (2023), but incorporates a set of modifications to fit our assumed penalized log-likelihood. Specifically, we exclude the additional linear regression term, model the scale parameter through splines rather than linear regression and introduce a penalty $\boldsymbol{\kappa}$ for the scale. Moreover, we extend the algorithm to the GAM setting and assume fixed penalty parameters $\boldsymbol{\alpha}$ and $\boldsymbol{\kappa}$.

Ferreira & Paula (2017) show that the skew-normal density from Equation (2) can be written in terms of a positive latent variable $V$ through

$$p(y \,|\, \xi, \omega, \lambda) = 2\phi\left(y; \xi, \omega^2\right) \int_0^{+\infty} \phi\left(v; \lambda(y - \xi), \omega\right) dv,$$

where $\phi(x; \mu, \sigma^2)$ denotes the $\mathcal{N}(\mu, \sigma^2)$ pdf evaluated at $x$. The joint pdf of $Y$ and $V$ is then given by

$$p(y, v \,|\, \xi, \omega, \lambda) = 2\phi\left(y; \xi, \omega\right) \phi\left(v; \lambda(y - \xi), \omega\right). \quad (5)$$

It follows that $V|Y = y \sim \text{TN}(\lambda(y-\xi), \omega^2, 0, +\infty)$, where TN denotes the truncated normal distribution with parameters $\lambda(y - \xi)$ and $\omega^2$ and support $(0, +\infty)$ (Johnson et al., 1995). Due to Equation (5), the complete data log-likelihood for $\boldsymbol{y}$ and the latent $\boldsymbol{v} = (v_1, \ldots, v_n)^\top$ satisfying the cause-effect model from Equation (3) is given by

$$\ell_c(\boldsymbol{\theta} \,|\, \boldsymbol{y}, \boldsymbol{v}) = \sum_{i=1}^n \left( \lambda \frac{y_i - \mathbf{n}_i^\top \boldsymbol{\psi}}{\exp(\mathbf{z}_i^\top \boldsymbol{\rho})} v_i - \mathbf{z}_i^\top \boldsymbol{\rho} - \frac{1}{2} \frac{v_i^2}{\exp(\mathbf{z}_i^\top \boldsymbol{\rho})} \right)$$
$$- \frac{1 + \lambda^2}{2} \sum_{i=1}^n \frac{(y_i - \mathbf{n}_i^\top \boldsymbol{\psi})^2}{\exp(\mathbf{z}_i^\top \boldsymbol{\rho})} - n \log(\pi).$$

Based on $\widehat{\boldsymbol{\theta}}^{(k)} = ((\widehat{\boldsymbol{\psi}}^{(k)})^\top, (\widehat{\boldsymbol{\rho}}^{(k)})^\top, \widehat{\lambda}^{(k)})^\top$, denoting the parameter estimates after $k$ iterations, further define

$$\widehat{\boldsymbol{v}}^{(k)} = (\widehat{v}_1^{(k)}, \ldots, \widehat{v}_n^{(k)})^\top, \qquad \widehat{v}_i^{(k)} = \mathbb{E}\left[V_i|Y_i, \widehat{\boldsymbol{\theta}}^{(k)}\right],$$

$$\widehat{\boldsymbol{v^2}}^{(k)} = (\widehat{v^2}_1^{(k)}, \ldots, \widehat{v^2}_n^{(k)})^\top, \quad \widehat{v^2}_i^{(k)} = \mathbb{E}\left[V_i^2|Y_i, \widehat{\boldsymbol{\theta}}^{(k)}\right].$$

Define $\boldsymbol{a} = \boldsymbol{y} - \mathbf{N}\boldsymbol{\psi}$. Then, given $\widehat{\boldsymbol{\theta}}^{(k)}$, the $Q$-function is

$$Q(\boldsymbol{\theta} \,|\, \widehat{\boldsymbol{\theta}}^{(k)}) = \mathbb{E}\left[\ell_c(\boldsymbol{\theta}|\boldsymbol{y}, \boldsymbol{v}) \,|\, \boldsymbol{y}, \widehat{\boldsymbol{\theta}}^{(k)}\right]$$

$$\propto \lambda \boldsymbol{a}^\top \mathbf{H}\widehat{\boldsymbol{v}}^{(k)} - \sum_{i=1}^n \mathbf{z}_i^\top \boldsymbol{\rho} - \frac{1}{2}\mathbf{1}_n^\top \mathbf{H}\widehat{\boldsymbol{v^2}}^{(k)}$$

$$- \frac{1 + \lambda^2}{2}\boldsymbol{a}^\top \mathbf{H}\boldsymbol{a},$$

where $\mathbf{H} = \text{diag}(\omega_1^{-2}, \ldots, \omega_n^{-2})$ with $\omega_i = \exp(0.5\mathbf{z}_i^\top \boldsymbol{\rho})$. The expectation is taken with respect to the conditional density $\boldsymbol{V}|\boldsymbol{Y} = \boldsymbol{y}$. In order to obtain the penalized maximum likelihood estimate w.r.t. Equation (4), the quantity

$$Q_p(\boldsymbol{\theta}|\boldsymbol{\theta}^{(k)}) = Q(\boldsymbol{\theta}|\boldsymbol{\theta}^{(k)}) - \frac{1}{2}\boldsymbol{\psi}^\top \mathbf{K}_{\boldsymbol{\alpha}}\boldsymbol{\psi} - \frac{1}{2}\boldsymbol{\rho}^\top \mathbf{M}_{\boldsymbol{\kappa}}\boldsymbol{\rho}$$

is maximized by iteratively performing the following E-step and CM-steps. For the second CM-step, there is no analytical solution, so that we use the CMA-ES for maximization.

**1) E-step:** Given $\boldsymbol{y}$ and the current estimates $\widehat{\boldsymbol{\theta}}^{(k)}$, compute the conditional expectations of the latent variables appearing in the $Q$-function:

$$\widehat{v}_i^{(k)} = \widehat{\lambda}^{(k)}e_i^{(k)} + \widehat{\omega}_i^{(k)}W_\Phi\left(\frac{\widehat{\lambda}^{(k)}e_i^{(k)}}{\widehat{\omega}_i^{(k)}}\right),$$

$$\widehat{v^2}_i^{(k)} = \left(\widehat{\lambda}^{(k)}e_i^{(k)}\right)^2 + \left(\widehat{\omega}_i^{(k)}\right)^2$$
$$+ \widehat{\lambda}^{(k)}e_i^{(k)}\widehat{\omega}_i^{(k)}W_\Phi\left(\frac{\widehat{\lambda}^{(k)}e_i^{(k)}}{\widehat{\omega}_i^{(k)}}\right),$$

where $e_i^{(k)} = y_i - \mathbf{n}_i^\top \widehat{\boldsymbol{\psi}}^{(k)}$ and $W_\Phi(u) = \phi(u)(\Phi(u))^{-1}$.

**2) CM-step 1:** Fixing $\widehat{\boldsymbol{\rho}}^{(k)}$, update $\widehat{\boldsymbol{\psi}}^{(k)}$ and $\widehat{\lambda}^{(k)}$ as

$$\widehat{\boldsymbol{\psi}}^{(k+1)} = \left(\mathbf{N}^\top \widehat{\mathbf{H}}^{(k)}\mathbf{N} + \frac{1}{1 + (\widehat{\lambda}^{(k)})^2}\mathbf{K}_{\boldsymbol{\alpha}}\right)^{-1} \mathbf{N}^\top \widehat{\mathbf{H}}^{(k)}$$

$$\cdot \left(\boldsymbol{y} - \frac{\widehat{\lambda}^{(k)}}{1 + (\widehat{\lambda}^{(k)})^2}\widehat{\boldsymbol{v}}^{(k)}\right),$$

$$\widehat{\lambda}^{(k+1)} = \frac{\left(\boldsymbol{y} - \mathbf{N}\widehat{\boldsymbol{\psi}}^{(k)}\right)^\top \widehat{\mathbf{H}}^{(k)}\widehat{\boldsymbol{v}}^{(k)}}{\left(\boldsymbol{y} - \mathbf{N}\widehat{\boldsymbol{\psi}}^{(k)}\right)^\top \widehat{\mathbf{H}}^{(k)}\left(\boldsymbol{y} - \mathbf{N}\widehat{\boldsymbol{\psi}}^{(k)}\right)}.$$

**3) CM-step 2:** Given $\widehat{\boldsymbol{\psi}}^{(k+1)}$ and $\widehat{\lambda}^{(k+1)}$, update $\widehat{\boldsymbol{\rho}}^{(k)}$ as

$$\widehat{\boldsymbol{\rho}}^{(k+1)} = \text{argmax}_{\boldsymbol{\rho}} \, Q_p(\boldsymbol{\rho} \,|\, \widehat{\boldsymbol{\psi}}^{(k+1)}, \widehat{\lambda}^{(k+1)}).$$

The algorithm stops if $||\widehat{\boldsymbol{\theta}}^{(k+\Delta)} - \widehat{\boldsymbol{\theta}}^{(k)}||_2 < 10^{-6}$ or after a specified maximum number of iterations. We set $\Delta = 25$.

# 5. Causal Discovery with SKEWD

Our goal is to recover the underlying causal graph $\mathcal{G}$ from iid observations of $d$ variables $Y^{(1)}, \ldots, Y^{(d)}$ generated by an LSNM. Therefore, SKEWD, short for *skewness-robust discovery*, fits skew-normal LSNMs based on Algorithm 1. We first cover cause-effect inference in the bivariate setting $d = 2$, adopting the notational convention $X := Y^{(1)}$ and $Y := Y^{(2)}$. In this case, the task reduces to distinguishing between two graphs $X \to Y$ and $Y \to X$, which SKEWD accomplishes through either independence testing or likelihood scoring, as common in prior works. We then extend the likelihood scoring variant to the multivariate setting.

## 5.1. Cause-Effect Inference

**Independence Testing**   Regression with subsequent independence testing (Peters et al., 2014, RESIT) is a widely used causal discovery approach. If the true graph is $X \to Y$, where $Y = f_{X \to Y}(X) + g_{X \to Y}(X)N_Y$, then $X$ and $N_Y$ are independent, and analogously, $Y$ and $N_X$ should be independent in graph $Y \to X$, where $X = f_{Y \to X}(Y) + g_{Y \to X}(Y)N_X$. We estimate the noise terms as follows:

$$\widehat{N}_Y = \frac{Y - \widehat{f}_{X \to Y}(X)}{\widehat{g}_{X \to Y}(X)} \text{ and } \widehat{N}_X = \frac{X - \widehat{f}_{Y \to X}(Y)}{\widehat{g}_{Y \to X}(Y)}.$$

Subsequently, independence tests are performed for the cause-noise pair in each direction. SKEWD uses the HSIC test (Gretton et al., 2005) and infers the direction with the higher $p$-value due to higher evidence for independence. Figure 1 illustrates the noise extraction in the true causal direction of a simulated data pair. The full example including the reverse direction is provided in Appendix B.

**Likelihood Scoring**   To perform cause-effect inference via likelihood scoring, SKEWD evaluates the joint log-likelihoods $\ell_{X \to Y}$ and $\ell_{Y \to X}$, where

$$\ell_{X \to Y}(\boldsymbol{x}, \boldsymbol{y}) = \log(p(\boldsymbol{x})) + \log(p_{X \to Y}(\boldsymbol{y}|\boldsymbol{x}))$$

and $\ell_{Y \to X}$ is given analogously. The direction with the higher estimated log-likelihood is inferred by SKEWD. An estimate for $p_{X \to Y}(\boldsymbol{y}|\boldsymbol{x})$, the conditional density of $\boldsymbol{Y}|\boldsymbol{X}$ in direction $X \to Y$, is obtained by estimating the corresponding skew-normal LSNM. SKEWD with likelihood scoring assumes that the observations of $X$ and $Y$ have been standardized to avoid potential biases due to scaling (Reisach et al., 2021). It further assumes that the marginal distributions of $X$ and $Y$ are normal distributions. As shown in Appendix A, in that case, the marginal log-likelihoods can be considered equal and hence be neglected. We also investigated nonparametric estimation of the marginal distributions via kernel density estimation (KDE) in an ablation study detailed in Appendix B. These initial results are encouraging, motivating further investigation of this topic.

## 5.2. Multivariate Extension

We generalize the likelihood-based variant (SKEWD-LL) to the multivariate setting, yielding SKEWD-MV, which is a score-based causal discovery method that infers the DAG with the highest estimated likelihood. Since the number of possible DAGs grows super-exponentially with $d$, SKEWD-MV first estimates the Markov equivalence class (MEC) $\mathcal{M}$ using the order-independent PC algorithm (Colombo & Maathuis, 2014) yielding a set of candidate DAGs. Assuming faithfulness, the true causal graph is contained in $\mathcal{M}$. The method then compares the likelihoods of all DAGs in the estimated MEC $\widehat{\mathcal{M}}$. By the Markov property, the log-likelihood associated with a DAG $\mathcal{G} = (\boldsymbol{V}, \boldsymbol{E})$ is

$$\ell(\mathcal{G}) = \sum_{Y \in \boldsymbol{V}} \ell(Y|\text{PA}_Y^{\mathcal{G}}),$$

where $\ell(Y|\text{PA}_Y^{\mathcal{G}})$ denotes the conditional log-likelihood of $Y$ given its parents.

To score all DAGs in $\widehat{\mathcal{M}}$, it is not necessary to evaluate the full log-likelihood. It suffices to compare the partial sum

$$\ell_u(\mathcal{G}) = \sum_{Y \in \boldsymbol{V}_u} \ell(Y|\text{PA}_Y^{\mathcal{G}}),$$

where $\boldsymbol{V}_u \subseteq \boldsymbol{V}$ denotes the set of nodes with at least one unoriented edge in $\widehat{\mathcal{M}}$. For nodes without unoriented edges, the likelihood contributions are equal across all DAGs in $\widehat{\mathcal{M}}$. We formalize this result in Proposition 5.1 and provide the proof in Appendix A. Consequently, SKEWD-MV selects the DAG $\mathcal{G}$ in $\widehat{\mathcal{M}}$ that maximizes $\ell_u$.

**Proposition 5.1.** *Let $\mathcal{M}$ be the MEC of the true causal graph and let $\mathcal{C}$ denote the set of DAGs in $\mathcal{M}$. Let $\boldsymbol{V}_u$ be the set of nodes with an undirected edge in $\mathcal{M}$. Then,*

$$argmax_{\mathcal{G} \in \mathcal{C}} \, \ell(\mathcal{G}) = argmax_{\mathcal{G} \in \mathcal{C}} \, \ell_u(\mathcal{G}).$$

In the computation of $\ell_u(\mathcal{G})$, the conditional likelihoods are estimated based on Algorithm 1 and the marginals are estimated via maximum likelihood under the Gaussian assumption. As in the bivariate case, the data is standardized prior to model fitting and we explore nonparametric marginal estimation via KDE in Appendix B. We provide the pseudo code for SKEWD-MV in Algorithm 2 in Appendix A. Under correct estimation of both $\widehat{\mathcal{M}}$ and the conditional likelihoods, Proposition 5.1 implies that SKEWD-MV recovers the true causal graph whenever it uniquely maximizes the likelihood—which holds except in pathological cases, as discussed below in Section 5.4.

## 5.3. Multivariate Scalability

The computational cost of SKEWD-MV is mainly driven by the need to fit a skew-normal model for each admissible

parent-child configuration (PCC) arising from undirected edges in the MEC. Disregarding the DAG property, the number of such configurations for a specific parent node is $2^v - 1$ given it has $v$ undirected edges. Therefore, the number of PCCs for a single parent node grows exponentially with $v$. Moreover, the higher the number of children in a PCC, the more parameters need to be estimated.

That said, scalability depends strongly on the structure of the MEC rather than only on the number of variables. If the MEC contains relatively few admissible PCCs, even larger graphs are practically manageable. For reference, in our experiments with $d = 10$ nodes evaluated in Section 6, the maximum number of PCCs was 40. Hence, we expect that SKEWD-MV is still applicable if the PCCs are in the order of 100. Consequently, the method is most suitable for smaller-scale problems, where it provides reliable inference as demonstrated in Section 6. A practical way to improve scalability is to omit the ECM step, which reduces runtime at the cost of some accuracy. As discussed in Appendix B, the heuristic optimization achieves sufficient performance with respect to bivariate likelihood scoring, suggesting it may be a reasonable trade-off in larger-scale problems.

### 5.4. Notes on Identifiability

The previously discussed approaches to cause-effect inference and causal graph learning through independence testing or likelihood scoring are valid only if the skew-normal LSNM is identifiable. Immer et al. (2023) and Strobl & Lasko (2023) proved identifiability of the causal graph under LSNMs for bivariate models, apart from highly constrained settings. Strobl & Lasko (2023) extended these results to the multivariate setting, where non-identifiability occurs only if for some child-parent pair $(X, Y)$ and some conditioning set $\text{PA}_Y^{\mathcal{G}} \setminus \{X\} \subseteq \boldsymbol{S} \subseteq \text{ND}_Y^{\mathcal{G}} \setminus \{X, Y\}$ a specific ordinary differential equation holds for every conditioning value $\boldsymbol{s}$ with $p(\boldsymbol{s}) > 0$. Here, $\text{ND}_Y^{\mathcal{G}}$ denotes the non-descendants of $Y$. For clarity, we restate their results in Appendix A.

**Limitations** While these general identification results have mild restrictions on the distribution for the cause and the noise, in practice, the estimation procedure can be misspecified. That is, SKEWD-IT could be misspecified if the conditional cannot be modeled with a skew-normal, or SKEWD-LL, under the Gaussianity assumption, could fail for non-Gaussian marginal distributions as discussed in our ablation experiment (Appendix B).

## 6. Experiments

### 6.1. Cause-Effect Inference

We evaluate SKEWD with likelihood scoring (SKEWD-LL) and independence testing (SKEWD-IT) on established bivariate causal discovery benchmarks and novel synthetic

LSNM datasets that exhibit varying degrees of skewness. We compare the performance in terms of accuracy and area under the decision rate curve (AUDRC) against different state-of-the-art (SOTA) methods for bivariate causal discovery. The AUDRC measures how well the certainty of an algorithm coincides with its accuracy (Marx & Vreeken, 2019; Immer et al., 2023). For SKEWD, higher certainty corresponds to a larger absolute difference in the log-likelihoods or the $p$-values from independence tests. The AUDRC is computed by sorting predictions from most to least certain and averaging the accuracies over the top $m$ predictions for $m = 1, \ldots, M$, where $M$ is the number of pairs. Let $\pi(i)$ be the index of the $i$-th most certain prediction, $f(\pi(i))$ the predicted direction and $t(\pi(i))$ the correct direction, then

$$\text{AUDRC} = \frac{1}{M} \sum_{m=1}^{M} \frac{1}{m} \sum_{i=1}^{m} \mathbb{1}[f(\pi(i)) = t(\pi(i))].$$

As with accuracy, the AUDRC is bounded by 0 and 1, with larger values reflecting better performance.

**Baselines** We include LOCI (Immer et al., 2023), ROCHE (Tran et al., 2024), GRCI (Strobl & Lasko, 2023) and QCCD (Tagasovska et al., 2020) as LSNM-specific approaches. For LOCI, which assumes Gaussian noise, we consider both log-likelihood and independence test versions, LOCI-LL and LOCI-IT, with neural network estimators. ROCHE assumes $t$-distributed noise, and GRCI extracts residuals via cross-validated regression (both are independence-test-based). QCCD computes scores via nonparametric quantile regression. As ANM-specific baselines, we include CAM (Bühlmann et al., 2014) and RESIT (Peters et al., 2014). Lastly, we consider IGCI (Janzing et al., 2012) and IGCI with prior standardization (IGCI-G).

**Skew-Noise Datasets** To create LSNM (and ANM) datasets with skewed noise, we base our approach on the data-generating mechanisms used by Tagasovska et al. (2020). That is, we generate ANMs of the form $Y = f(X) + \varepsilon$ and LSNMs of the form $Y = f(X) + g(X)\varepsilon$, where $f, g$ are randomly sampled sigmoidal functions as in the original ANs and LSs datasets. Such injective functions pose a more challenging cause-effect inference than randomly generated potentially non-invertible functions. The cause is hierarchically sampled as $X \sim \mathcal{N}(0, \sigma^2)$, where $\sigma^2 \sim U[1, 2]$. For noise $\varepsilon$, we differentiate five cases. We consider two skew-normal settings, with $\varepsilon \sim \text{SN}(0, 1, \lambda)$ where $\lambda \in \{-2, 20\}$. Additionally, we consider three parameterizations of the generalized normal (GNO) distribution (Hosking, 1986) with $\varepsilon \sim \text{GNO}(0, 1, k)$, where the shape parameter $k$ is varied between $0.15$, $-0.31$, and $-0.5$. The GNO distribution is a reparametrized three-parameter lognormal distribution (Kozlov & Maysuradze, 2019) that generalizes the Gaussian. See Appendix A for more details.

For each case, we sample 100 pairs with $n = 1000$ observa-

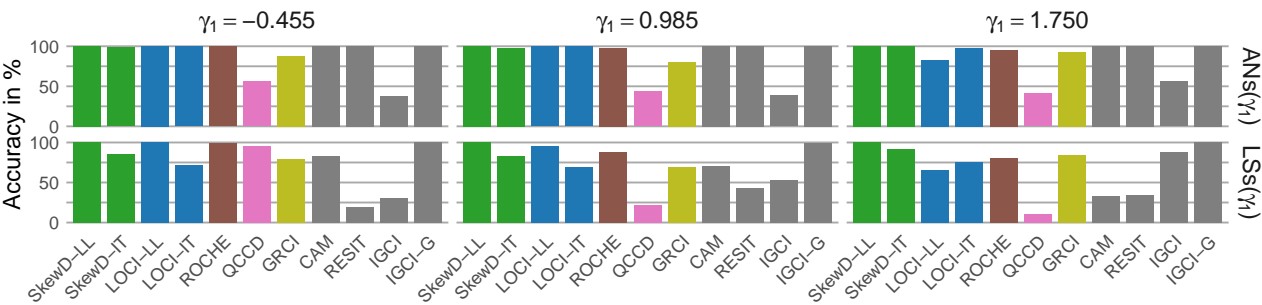

*Figure 2.* Accuracy in % for SKEWD and the baselines (approaches not developed for LSNMs in gray) on the proposed ANs and LSs datasets with skewness levels $-0.455$, $0.985$ and $1.750$. SKEWD consistently performs well, whereas the likelihood-based variant shows the strongest performance. IGCI-G while strongly misspecified seems to be biased toward the correct direction.

tions each across the ANs and LSs setting. For our analysis, we combine the skew-normal datasets with $\lambda = -2$ and GNO datasets with $k = 0.15$ into one dataset class, since they both exhibit roughly the same skewness ($\gamma_1 \approx -0.455$). Similarly, we group together the datasets with $\lambda = 20$ and $k = -0.31$, both expressing a skewness of $\gamma_1 \approx 0.985$. The $\text{GNO}(0, 1, -0.5)$ distribution yields a higher skewness $\gamma_1 \approx 1.750$ than what can be modeled by the skew-normal distribution. For reference, the exponential distribution always exhibits a skewness of 2. We denote the resulting datasets in correspondence to the specific skewness level $\gamma_1$ as ANs($\gamma_1$) and LSs($\gamma_1$). Scatter plots of data pairs from each generated dataset are provided in Appendix C.

By including the GNO parametrizations in the skew-noise benchmark, we additionally evaluate robustness under substantial deviations from the skew-normal assumption. Beyond the increased skewness level considered for $k = -0.5$, the GNO also exhibits different tail behavior than the SN, having a truncated tail and potentially heavier tails. For $k = -0.31$ and $k = -0.5$, the excess kurtosis is 1.77 and 5.90, respectively, with the latter comparable to a more heavy-tailed exponential or t-distribution with 5 degrees of freedom. In comparison, the skew-normal can only exhibit a maximum excess kurtosis of 0.87 (Azzalini, 2013).

**Established Benchmark Datasets** We consider several established benchmarks (a total of 13) to investigate the general applicability of SKEWD. These include the original AN, ANs, LS, LSs and MNU dataset featuring multiplicative noise from Tagasovska et al. (2020). We also consider the synthetic SIM, SIMc (includes confounding), SIMln (low noise), SIMG (near Gaussian ANM) datasets as well as the 99 continuous bivariate datasets real-world Tübingen benchmark (Mooij et al., 2016). Lastly, we include the Cha (used in the cause-effect pair challenge), Net (generated from random neural networks) and Multi datasets (generated from polynomial mechanisms) (Guyon et al., 2019) .

**Results Under Skewed Noise** In Figure 2, we plot the accuracy of SKEWD and all baselines when applied to the

proposed skew-noise datasets. We supplement the exact values for both accuracy and AUDRC in Appendix B. As the results for both metrics are consistent, we focus the discussion on accuracy. Notably, SKEWD-LL is the only method to achieve perfect accuracy across all cases and outperforms SKEWD-IT, especially in the LSs setting. IGCI-G demonstrates comparable performance, making only a single error across all datasets, although IGCI-G is highly misspecified, since it assumes almost deterministic functions. Among the LSNM-specific methods, ROCHE is the most reliable aside from SKEWD. However, its performance degrades in increasingly skewed LSNM settings, dropping to 81% accuracy for LSs(1.750). Similarly, both versions of LOCI and QCCD exhibit decreasing accuracy under increased skewness with QCCD becoming severely biased towards the false direction in the LSs case. The ANM-specific methods CAM and RESIT show robustness towards skewed ANMs, but their accuracy drops for LSNMs.

We also evaluated CAREFL (Khemakhem et al., 2021) as another nonparametric method based on normalizing flows, with the default cause-effect pair hyperparameter configuration. Similar to QCCD, its accuracy on the LSNM datasets decreases with increasing skewness (0.90, 0.71, 0.66). Since CAREFL did not perform competitively on the established Tübingen and SIM benchmarks, achieving accuracies no higher than 0.44, we omitted it from the main comparison.

**Results for Established Benchmarks** We summarize the performances of all baselines in terms of mean accuracy and AUDRC across all 13 established benchmarks in Table 1 including the respective standard deviations. The results demonstrate that SKEWD performs competitively. SKEWD-IT ranks second with a difference of 2.67% in accuracy to ROCHE, which is expected to have an advantage to our approach when distributions are heavy-tailed. Compared to LOCI, the larger model class that we consider shows a slight advantage on the established benchmarks and a strong advantage on the proposed skewed benchmarks. Further, our results confirm the observations made by previous ap-

*Table 1.* Mean and standard deviation of accuracy, AUDRC over all 13 established benchmarks. Top two are bolded.

| METHOD | ACCURACY (%) ↑ | AUDRC (%) ↑ |
|---|---|---|
| SKEWD-LL | $77.43 \pm 20.56$ | $84.77 \pm 17.87$ |
| SKEWD-IT | $\mathbf{85.02 \pm 9.71}$ | $\mathbf{91.65 \pm 9.32}$ |
| LOCI-LL | $77.08 \pm 22.22$ | $84.65 \pm 18.54$ |
| LOCI-IT | $84.09 \pm 12.20$ | $89.51 \pm 12.74$ |
| ROCHE | $\mathbf{87.69 \pm 10.65}$ | $\mathbf{92.90 \pm 8.40}$ |
| QCCD | $78.21 \pm 17.29$ | $85.77 \pm 14.26$ |
| GRCI | $83.20 \pm 9.82$ | $89.77 \pm 10.54$ |
| CAM | $72.42 \pm 22.10$ | $73.85 \pm 22.87$ |
| RESIT | $61.85 \pm 32.62$ | $66.25 \pm 31.68$ |
| IGCI | $46.36 \pm 20.66$ | $49.60 \pm 24.74$ |
| IGCI-G | $73.69 \pm 23.92$ | $77.21 \pm 25.04$ |

proaches (Immer et al., 2023; Sun & Schulte, 2023) that methods based on subsequent independence testing (in our case SKEWD-IT) are more robust across various settings.

In addition to the above analysis, we provide the individual results for each dataset in Appendix B. Most notably, with 88% accuracy on SIMG, SKEWD-LL outperforms all remaining methods in terms of accuracy by at least 5%. Over the original AN, ANs, LS, and LSs datasets with normally distributed noise, SKEWD-LL scores perfect accuracy demonstrating its effectiveness under symmetric and skewed noise. We also compute the mean accuracy across all 19 datasets (including our novel datasets), for which SKEWD-LL improves to 84.56%, and SKEWD-IT to 87.46%.

### 6.2. Causal Discovery

As a proof of concept, we evaluate our multivariate approach (SKEWD-MV) against SOTA causal discovery methods on synthetic data generated by skew-normal LSNM graphs with $d = 6$ and $d = 10$ nodes. For each $d$, 100 datasets of size $n = 1000$ are generated based on Erdős-Rényi graphs with $b = d$ edges (Erdős & Rényi, 1960). The functions $f^{(j)}$ and $g^{(j)}$ are constructed as sums of random sigmoidal functions over the causes of $Y^{(j)}$, whereas $N_{Y^{(j)}} \sim SN(0, 1, 0.985)$. Source nodes are generated via $Y^{(j)} \sim U[-3.5, 3.5]$.

We evaluate the performance in terms of structural Hamming distance (SHD), counting reversed edges as two errors, and structural intervention distance (SID) (Peters & Bühlmann, 2015). We include HOST (Duong & Nguyen, 2023), GraN-DAG++ (Lachapelle et al., 2020) and CAM (Bühlmann et al., 2014) as baselines. While HOST and GraN-DAG++ model LSNMs, CAM assumes ANMs. We summarize the results in Table 2, where we also include SKEWD-MV-O representing SKEWD-MV using the oracle MEC instead of the estimate from the PC algorithm. SKEWD-MV substantially outperforms all baselines in both metrics. SKEWD-MV-O achieves near-perfect performance, correctly orienting 97.28% ($d = 6$) and 99.43% ($d = 10$) of edges. These results demonstrate the effectiveness of

*Table 2.* Mean SHD and SID on synthetic data. Top two are bolded. For GraN-DAG++, 5 error-terminated runs were excluded.

| METHOD | $d = 6$ | | $d = 10$ | |
|---|---|---|---|---|
| | SHD ↓ | SID ↓ | SHD ↓ | SID ↓ |
| SKEWD-MV | $\mathbf{2.27}$ | $\mathbf{4.35}$ | $\mathbf{4.72}$ | $\mathbf{14.97}$ |
| SKEWD-MV-O | $\mathbf{0.14}$ | $\mathbf{0.23}$ | $\mathbf{0.04}$ | $\mathbf{0.10}$ |
| HOST | 8.90 | 12.84 | 21.21 | 36.10 |
| GRAN-DAG++ | 11.30 | 12.32 | 24.67 | 32.26 |
| CAM | 13.52 | 9.23 | 42.08 | 19.33 |

SKEWD-MV on small to moderately sized graphs. However, since the number of required skew-normal regressions scales with the number of edges, computational scalability may become a limitation for large, dense graphs. We additionally report precision, recall, and F1-score for these experiments in Appendix B where SKEWD-MV also outperforms all remaining baselines across all three metrics.

## 7. Conclusion

In this paper, we focused on bivariate causal discovery under skewed noise distributions. We proposed a set of novel datasets, based on which we demonstrated that state-of-the-art (SOTA) methods for location-scale noise models become unreliable under increasingly skewed noise. To address this issue we proposed SKEWD, which performs cause-effect inference based on independence testing or likelihood scoring under the assumption of skew-normally distributed noise. For parameter estimation, we devised a novel estimation approach for skew-normal LSNMs based on an ECM algorithm. Our results show that SKEWD is, in contrast to SOTA, robust with respect to higher amounts of skewness while performing on par with SOTA on established benchmarks. We further study a multivariate variant of SKEWD, demonstrating its applicability beyond the bivariate setting.

**Limitations and Future Work**    Natural extensions of our work would be to consider a multivariate extension of the independence-based cause-effect inference, enabling interaction effects in the multivariate setting, or further investigating nonparametric marginals beyond kernel density estimation to lift the Gaussianity assumption for the likelihood-based approach.

## Acknowledgements

The authors would like to thank Clécio da Silva Ferreira for supporting this work by generously providing code and valuable insights. They also gratefully acknowledge the computing time provided on the Linux HPC cluster at Technical University Dortmund (LiDO3), partially funded in the course of the Large-Scale Equipment Initiative by the Deutsche Forschungsgemeinschaft (DFG, German Research Foundation) as project 271512359.

## Impact Statement

This work contributes to improving causal discovery under skewed noise, which may increase the reliability of causal analyses in real-world applications. At the same time, causal discovery methods can introduce risks in high-stakes domains, since incorrect causal conclusions may result in harmful interventions. Importantly, the identifiability of causal directions in our framework only holds under the stated assumptions including the absence of unobserved confounding. Violations of these assumptions may lead to invalid causal conclusions.

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

# A. Technical Details

## A.1. Skew-Distributions

In this section, we present more details on the considered skewed distributions, the skew-normal and the generalized normal. This includes their pdf, expected value, variance and skewness.

**Skew-Normal** As discussed in Section 3, the random variable $X$ follows a univariate skew-normal distribution (Azzalini, 1985) with parameters $\xi \in \mathbb{R}$ (location), $\omega > 0$ (scale), and $\lambda \in \mathbb{R}$ (shape), $X \sim \text{SN}(\xi, \omega, \lambda)$, if its pdf is given by

$$p(x) = \frac{2}{\omega} \phi\left(\frac{x - \xi}{\omega}\right) \Phi\left(\lambda \frac{x - \xi}{\omega}\right), \quad x \in \mathbb{R}.$$

Define $b = \sqrt{2/\pi}$ and $\delta = \lambda/\sqrt{(1 + \lambda^2)}$. Then, the expected value, the variance and the skewness of $X \sim \text{SN}(\xi, \omega, \lambda)$ are given by

$$\mathbb{E}[X] = \xi + \omega b \delta, \quad \text{Var}(X) = \omega^2(1 - b^2 \delta^2), \quad \text{and} \quad \gamma_1 = \frac{4 - \pi}{2} \frac{b^3 \delta^3}{(1 - b^2 \delta^2)^{3/2}}. \tag{6}$$

The skew-normal distribution is right-skewed if $\lambda > 0$ and left-skewed if $\lambda < 0$ and its skewness is bounded by the interval $(-\gamma_1^{\max}, \gamma_1^{\max})$, where $\gamma_1^{\max} \approx 0.9953$. If $\lambda = 0$, the distribution corresponds to a normal distribution with mean $\xi$ and variance $\omega^2$.

In the skew-normal LSNM defined in Section 3, the cause given its effects is modeled as $Y|\boldsymbol{X} = \boldsymbol{x} \sim \text{SN}(f(\boldsymbol{x}), g(\boldsymbol{x}), \lambda)$. That is, we model the location and scale parameter through functions $f$ and $g$ as defined in Equation (3). We note that, unless $\lambda = 0$, it follows from Equation (6) that the modeling functions $f$ and $g$ do not correspond to the mean and the standard deviation of the cause given its effects. Given estimates $\widehat{f}, \widehat{g}$ and $\widehat{\lambda}$, the mean and variance estimates are recovered as

$$\widehat{\mathbb{E}}[Y|\boldsymbol{X} = \boldsymbol{x}] = \widehat{f}(\boldsymbol{x}) + \widehat{g}(\boldsymbol{x})b\widehat{\delta} \quad \text{and} \quad \widehat{\text{Var}}(Y|\boldsymbol{X} = \boldsymbol{x}) = (\widehat{g}(\boldsymbol{x}))^2(1 - b^2\widehat{\delta}^2),$$

where $\widehat{\delta} = \widehat{\lambda}/\sqrt{(1 + \widehat{\lambda}^2)}$.

**Generalized Normal** The generalized normal (GNO) distribution (Hosking, 1986) is a reparametrized version of the three-parameter lognormal distribution (see for instance Kozlov & Maysuradze (2019)). The random variable $X$ follows a GNO distribution, $X \sim \text{GNO}(\xi, \alpha, k)$, if its pdf is given by

$$p(x) = \frac{1}{\sqrt{2\pi}} \frac{1}{\alpha} \exp(ky - 0.5y^2), \quad \text{where } y = \begin{cases} -\frac{1}{k}\log(1 - k\frac{x-\xi}{\alpha}) & , k \neq 0 \\ \frac{x-\xi}{\alpha} & , k = 0 \end{cases}$$

with support $\xi + \frac{\alpha}{k} \leq x < \infty$ if $k < 0$, $-\infty < x < \infty$ if $k = 0$, and $-\infty < x \leq \xi + \frac{\alpha}{k}$ if $k > 0$. It follows that the GNO distribution with $k = 0$, $X \sim \text{GNO}(\xi, \alpha, 0)$, corresponds to the regular normal distribution with mean $\xi$ and variance $\alpha^2$. For $k \neq 0$, the mean and variance are given by

$$\mathbb{E}[X] = \xi - \frac{\alpha}{k}\left(\exp(0.5k^2) - 1\right), \quad \text{Var}(X) = \frac{\alpha^2}{\kappa^2}\exp(k^2)\left(\exp(k^2) - 1\right)$$

and the skewness corresponds to

$$\gamma_1 = \text{sign}(k)\frac{3\exp(k^2) - \exp(3k^2) - 2}{(\exp(k^2) - 1)^{3/2}}.$$

When $k < 0$ the GNO distribution has a lower bound and exhibits positive skewness, whereas when $k > 0$ the GNO distribution has an upper bound and exhibits negative skewness.

## A.2. Details on the Skew-Normal Model

In this subsection, we provide the explicit formulation of the skew-normal introduced in Section 3. Given causes $\boldsymbol{X} = (X^{(1)}, \dots, X^{(s)})^\top$, the effect $Y$ is modeled as $Y|\boldsymbol{X} = \boldsymbol{x} \sim \text{SN}(f(\boldsymbol{x}), g(\boldsymbol{x}), \lambda)$, where both functions $f$ and $g$ are modeled

using generalized additive models (GAMs):

$$f(\boldsymbol{x}) = \sum_{t=1}^{s} f_t(x^{(t)}), \quad g(\boldsymbol{x}) = \exp\left(\sum_{t=1}^{s} g_t(x^{(t)})\right).$$

Each covariate-specific component function $f_t$ and $g_t$ is modeled by an individual B-spline.

For observation $i$ with covariate vector $\boldsymbol{x}_i$, $i = 1, \dots, n$, the $t$-th components are given by

$$f_t(x_i^{(t)}) = \widetilde{\mathbf{n}}_{t;i}^{\top} \widetilde{\boldsymbol{\psi}}_t, \quad g_t(x_i^{(t)}) = \widetilde{\mathbf{z}}_{t;i}^{\top} \widetilde{\boldsymbol{\rho}}_t,$$

where $\widetilde{\mathbf{n}}_{t;i}^{\top} \in \mathbb{R}^{1 \times \widetilde{q}}$, $\widetilde{\mathbf{z}}_{t;i}^{\top} \in \mathbb{R}^{1 \times \widetilde{p}}$ correspond to the $i$-th rows of the spline design matrices $\mathbf{N}_t \in \mathbb{R}^{n \times \widetilde{q}}$ and $\mathbf{Z}_t \in \mathbb{R}^{n \times \widetilde{p}}$. The corresponding unknown coefficient vectors are $\widetilde{\boldsymbol{\psi}}_t = (\widetilde{\psi}_{t;1}, \dots, \widetilde{\psi}_{t;\widetilde{q}})^{\top} \in \mathbb{R}^{\widetilde{q}}$ and $\widetilde{\boldsymbol{\rho}}_t = (\widetilde{\rho}_{t;1}, \dots, \widetilde{\rho}_{t;\widetilde{p}})^{\top} \in \mathbb{R}^{\widetilde{p}}$. Consequently, the spline basis for each covariate has identical size $\widetilde{q}$ for $f$ and $\widetilde{p}$ for $g$.

By setting

$$\mathbf{N} = (\mathbf{N}_1, \dots, \mathbf{N}_s) \in \mathbb{R}^{n \times q}, \quad \mathbf{Z} = (\mathbf{Z}_1, \dots, \mathbf{Z}_s) \in \mathbb{R}^{n \times p}, \quad \boldsymbol{\psi} = \begin{pmatrix} \widetilde{\boldsymbol{\psi}}_1 \\ \vdots \\ \widetilde{\boldsymbol{\psi}}_s \end{pmatrix} \in \mathbb{R}^q, \quad \boldsymbol{\rho} = \begin{pmatrix} \widetilde{\boldsymbol{\rho}}_1 \\ \vdots \\ \widetilde{\boldsymbol{\rho}}_s \end{pmatrix} \in \mathbb{R}^p,$$

where $q = s \cdot \widetilde{q}$ and $p = s \cdot \widetilde{p}$, the cause-effect model can be written as

$$Y_i = f(\boldsymbol{x}_i) + g(\boldsymbol{x}_i) N_{Y_i} = \mathbf{n}_i^{\top} \boldsymbol{\psi} + \exp\left(0.5 \mathbf{z}_i^{\top} \boldsymbol{\rho}\right) N_{Y_i},$$

which coincides with Equation (3). Here, $\mathbf{n}_i^{\top}$ and $\mathbf{z}_i^{\top}$ correspond to the $i$-th row of the matrices $\mathbf{N}$ and $\mathbf{Z}$, respectively.

The B-spline penalties required for specifying the penalized likelihood are obtained as follows. Let

$$\mathbf{D}_r = \begin{bmatrix} 1 & -2 & 1 & 0 & \cdots & 0 \\ 0 & 1 & -2 & 1 & \cdots & 0 \\ \vdots & \ddots & \ddots & \ddots & \ddots & \vdots \\ 0 & \cdots & 0 & 1 & -2 & 1 \end{bmatrix} \in \mathbb{R}^{(r-2) \times r}$$

define the second-order difference matrix with respect to dimensionality $r$. Then, the penalty matrices for $\widetilde{\boldsymbol{\psi}}_t$ and $\widetilde{\boldsymbol{\rho}}_t$ are given by $\mathbf{K} = \mathbf{D}_{\widetilde{q}}^{\top} \mathbf{D}_{\widetilde{q}}$ and $\mathbf{M} = \mathbf{D}_{\widetilde{p}}^{\top} \mathbf{D}_{\widetilde{p}}$, respectively. Given penalty strength $\alpha_t$, the resulting penalty on $\widetilde{\boldsymbol{\psi}}_t$ is given by

$$\frac{1}{2} \alpha_t \widetilde{\boldsymbol{\psi}}_t^{\top} \mathbf{K} \widetilde{\boldsymbol{\psi}}_t = \frac{1}{2} \alpha_t \sum_{j=3}^{\widetilde{q}} (\widetilde{\psi}_{t;j} - 2\widetilde{\psi}_{t;j-1} + \widetilde{\psi}_{t;j-2})^2$$

and analogously for $\widetilde{\boldsymbol{\rho}}_t$ using $\mathbf{M}$ given $\kappa_t$.

Let

$$\mathbf{K}_{\boldsymbol{\alpha}} = \begin{pmatrix} \alpha_1 \mathbf{K} & 0 & \cdots & 0 \\ 0 & \alpha_2 \mathbf{K} & \cdots & 0 \\ \vdots & \vdots & \ddots & \vdots \\ 0 & \cdots & 0 & \alpha_s \mathbf{K} \end{pmatrix}, \quad \mathbf{M}_{\boldsymbol{\kappa}} = \begin{pmatrix} \kappa_1 \mathbf{M} & 0 & \cdots & 0 \\ 0 & \kappa_2 \mathbf{M} & \cdots & 0 \\ \vdots & \vdots & \ddots & \vdots \\ 0 & \cdots & 0 & \kappa_s \mathbf{M} \end{pmatrix},$$

The penalties on $\boldsymbol{\psi}$ and $\boldsymbol{\rho}$ are then represented by

$$\frac{1}{2} \boldsymbol{\psi}^{\top} \mathbf{K}_{\boldsymbol{\alpha}} \boldsymbol{\psi} = \frac{1}{2} \sum_{t=1}^{s} \widetilde{\boldsymbol{\psi}}_t^{\top} \alpha_t \mathbf{K} \widetilde{\boldsymbol{\psi}}_t, \quad \frac{1}{2} \boldsymbol{\rho}^{\top} \mathbf{M}_{\boldsymbol{\kappa}} \boldsymbol{\rho} = \frac{1}{2} \sum_{t=1}^{s} \widetilde{\boldsymbol{\rho}}_t^{\top} \kappa_t \mathbf{M} \widetilde{\boldsymbol{\rho}}_t.$$

Let $\ell(\boldsymbol{\theta})$ denote the observed log-likelihood of the above cause-effect model w.r.t. $\boldsymbol{\theta} = (\boldsymbol{\psi}^{\top}, \boldsymbol{\rho}^{\top}, \lambda)^{\top}$. The penalized log-likelihood is then given by

$$\ell_p(\boldsymbol{\theta}, \boldsymbol{\alpha}, \boldsymbol{\kappa}) = \ell(\boldsymbol{\theta}) - \frac{1}{2} \boldsymbol{\psi}^{\top} \mathbf{K}_{\boldsymbol{\alpha}} \boldsymbol{\psi} - \frac{1}{2} \boldsymbol{\rho}^{\top} \mathbf{M}_{\boldsymbol{\kappa}} \boldsymbol{\rho}. \tag{7}$$

which coincides with Equation (4).

## A.3. Neglection of Marginal Log-Likelihoods

When performing likelihood scoring with SKEWD, it is assumed that the cause is normally distributed and that the observations have been standardized for model estimation. We argue that in the bivariate case, the marginal log-likelihoods $p(\boldsymbol{x})$, $p(\boldsymbol{y})$ of the hypothesized causes can be neglected. By Proposition A.1 stated below, we can treat the standardized variables as approximately standard normal in large samples and thus model the marginal likelihoods via standard normal distributions. Let $x_1, \ldots, x_n$ denote the observed standardized causes with respect to SCM $X \to Y$. Then,

$$\log(p_{X \to Y}(\boldsymbol{x})) = \sum_{i=1}^{n} \log(\phi(x_i)) = -\frac{n}{2}\log(2\pi) - \frac{1}{2}\sum_{i=1}^{n} x_i^2 = -\frac{n}{2}\log(2\pi) - \frac{n-1}{2},$$

where the last equality stems from the fact that the sum of $n$ squared standardized observations is equal to $n-1$. To show this, let $v_1, \ldots, v_n$ denote the non-standardized observed causes and define $\bar{v} = \frac{1}{n}\sum_{j=1}^{n} v_i$ and $s = \sqrt{\frac{1}{n-1}\sum_{i=1}^{n}(v_i - \bar{v})^2}$. Then,

$$\sum_{i=1}^{n} x_i^2 = \sum_{i=1}^{n} \frac{(v_i - \bar{v})^2}{s^2} = \frac{\sum_{i=1}^{n}(v_i - \bar{v})^2}{\frac{1}{n-1}\sum_{i=1}^{n}(v_i - \bar{v})^2} = n-1.$$

Analogously, we can approximate $\log(p_{Y \to X}(\boldsymbol{y}))$ and due to the equality of the marginal log-likelihoods neglect these two terms when comparing the model log-likelihoods

$$\ell_{X \to Y}(\boldsymbol{x}, \boldsymbol{y}) = \log(p(\boldsymbol{x})) + \log(p_{X \to Y}(\boldsymbol{y}|\boldsymbol{x})) \gtrless \log(p(\boldsymbol{y})) + \log(p_{Y \to X}(\boldsymbol{x}|\boldsymbol{y})) = \ell_{Y \to X}(\boldsymbol{x}, \boldsymbol{y}).$$

**Proposition A.1.** *Let $Z_1, \ldots, Z_n \overset{iid}{\sim} \mathcal{N}(\mu, \sigma^2)$. Further define the standardized random variables*

$$\widetilde{Z}_i = \frac{Z_i - \bar{Z}_n}{S_n},$$

*where*

$$\bar{Z}_n = \frac{1}{n}\sum_{i=1}^{n} Z_i \quad and \quad S_n = \sqrt{\frac{1}{n-1}\sum_{i=1}^{n}(Z_i - \bar{Z}_n)^2}.$$

*Then, for $i = 1, \ldots, n$,*

$$\widetilde{Z}_i \xrightarrow{a.s.} \frac{Z_i - \mu}{\sigma} \sim \mathcal{N}(0, 1).$$

**Proof.** It is a well-known result in asymptotic theory that $\bar{Z}_n \xrightarrow{a.s.} \mu$ and $S_n \xrightarrow{a.s.} \sigma$. Thus,

$$\left(Z_i, \ \bar{Z}_n, \ S_n\right)^\top \xrightarrow{a.s.} \left(Z_i, \ \mu, \ \sigma\right)^\top$$

and applying the Continuous Mapping Theorem yields that

$$\widetilde{Z}_i = \frac{Z_i - \bar{Z}_n}{S_n} \xrightarrow{a.s.} \frac{Z_i - \mu}{\sigma}.$$

From the property of normal distributions, it follows that $\frac{Z_i - \mu}{\sigma} \sim \mathcal{N}(0, 1)$. $\qquad\square$

## A.4. Proof of Proposition 5.1

**Proposition 5.1 (restated).** *Let $\mathcal{M}$ be the MEC of the true causal graph and let $\mathcal{C}$ denote the set of DAGs in $\mathcal{M}$. Let $\boldsymbol{V}_u$ be the set of nodes with an undirected edge in $\mathcal{M}$. Then,*

$$argmax_{\mathcal{G} \in \mathcal{C}} \ell(\mathcal{G}) = argmax_{\mathcal{G} \in \mathcal{C}} \ell_u(\mathcal{G}).$$

**Proof.** The key observation is that for any two DAGs $\mathcal{G}_1, \mathcal{G}_2$ in the Markov equivalence class $\mathcal{M}$, the parent set of any node $Y$ without an undirected edge in $\mathcal{M}$ is identical. Formally,

$$\forall \mathcal{G}_1, \mathcal{G}_2 \in \mathcal{C}, \forall Y \in \boldsymbol{V} \setminus \boldsymbol{V}_u : \quad \text{PA}_Y^{\mathcal{G}_1} = \text{PA}_Y^{\mathcal{G}_2}.$$

To verify this, suppose for contradiction that the parent sets differ for some node $Y \in \boldsymbol{V} \setminus \boldsymbol{V}_u$. Then, there exists a node $X$ such that $X \in \text{PA}_Y^{\mathcal{G}_1}$ but $X \notin \text{PA}_Y^{\mathcal{G}_2}$ without loss of generality. Since all DAGs in $\mathcal{M}$ share the same skeleton, the edge between $X$ and $Y$ must also be present in $\mathcal{G}_2$. Hence, $Y \in \text{PA}_X^{\mathcal{G}_2}$, i.e. the edge between $X$ and $Y$ is reversed in $\mathcal{G}_2$, implying that the edge between $X$ and $Y$ is unoriented in $\mathcal{M}$. This contradicts the assumption that $Y \notin \boldsymbol{V}_u$.

It follows that the log-likelihood contributions $\ell(Y|\text{PA}_Y^{\mathcal{G}})$ are identical for all $Y \in \boldsymbol{V} \setminus \boldsymbol{V}_u$ and all $\mathcal{G} \in \mathcal{C}$. Therefore,

$$
\begin{aligned}
\text{argmax}_{\mathcal{G} \in \mathcal{C}} \, \ell(\mathcal{G}) &= \text{argmax}_{\mathcal{G} \in \mathcal{C}} \sum_{Y \in \boldsymbol{V}} \ell(Y|\text{PA}_Y^{\mathcal{G}}) \\
&= \text{argmax}_{\mathcal{G} \in \mathcal{C}} \left( \sum_{Y \in \boldsymbol{V}_u} \ell(Y|\text{PA}_Y^{\mathcal{G}}) + \sum_{Y \notin \boldsymbol{V}_u} \ell(Y|\text{PA}_Y^{\mathcal{G}}) \right) \\
&= \text{argmax}_{\mathcal{G} \in \mathcal{C}} \sum_{Y \in \boldsymbol{V}_u} \ell(Y|\text{PA}_Y^{\mathcal{G}}) \\
&= \text{argmax}_{\mathcal{G} \in \mathcal{C}} \, \ell_u(\mathcal{G}),
\end{aligned}
$$

where the equality follows from the fact that the second sum is constant over $\mathcal{C}$. $\qquad\square$

### A.5. SKEWD-MV Algorithm

In Algorithm 2 we provide the pseudo code for SKEWD-MV. We note that the described algorithm is simplified compared to the actual implementation which includes parallelization and prevents multiple estimation of the same conditionals. Within the PC algorithm, we use a kernel-based conditional independence test (Zhang et al., 2011).

---

**Algorithm 2** SKEWD-MV

---

1: **Input:** Data matrix $\boldsymbol{X} \in \mathbb{R}^{n \times d}$
2: Standardize the columns of $\boldsymbol{X}$
3: Estimate the Markov equivalence class $\mathcal{M} = \text{PC}(\boldsymbol{X})$
4: Define $\mathcal{I}$ as the set of nodes with at least one unoriented edge
5: Define $\mathcal{C}$ as the set of all DAGs in $\mathcal{M}$
6: Initialize scores $\ell_u(\mathcal{G}) \leftarrow 0$ for all $\mathcal{G} \in \mathcal{C}$
7: **for** each $Y \in \mathcal{I}$ **do**
8:     **for** each $\mathcal{G} \in \mathcal{C}$ **do**
9:         **if** $\text{PA}_Y^{\mathcal{G}} \neq \emptyset$ **then**
10:             Estimate $\widehat{\ell}(Y \mid \text{PA}_Y^{\mathcal{G}})$ using Algorithm 1
11:         **else**
12:             Estimate $\widehat{\ell}(Y|\text{PA}_Y^{\mathcal{G}}) = \widehat{\ell}(Y)$
13:         **end if**
14:         $\widehat{\ell}_u(\mathcal{G}) \leftarrow \widehat{\ell}_u(\mathcal{G}) + \widehat{\ell}(Y \mid \text{PA}_Y^{\mathcal{G}})$
15:     **end for**
16: **end for**
17: $\widehat{\mathcal{G}} \leftarrow \arg\max_{\mathcal{G} \in \mathcal{C}} \widehat{\ell}_u(\mathcal{G})$
18: **Return:** Estimated DAG $\widehat{\mathcal{G}}$

---

### A.6. Identifiability in Location-Scale Noise Models

In the following, we restate the theorem on the identification of location-scale noise models for bivariate models, that has been derived independently by Immer et al. (2023) and Strobl & Lasko (2023), for completeness. The authors derive a partial differential equation (PDE) which is required to be zero for all non-identifiable cases. They argue that only pathological cases which solve this differential equation exist.

**Theorem A.2** (Immer et al. (2023)). *Assume the data is such that a location-scale noise model can be fit in both directions, i.e.,*

$$Y = f(X) + g(X) N_Y, \quad X \perp\!\!\!\perp N_Y$$
$$X = h(Y) + k(Y) N_X, \quad Y \perp\!\!\!\perp N_X.$$

*Let $\nu_1(\cdot)$ and $\nu_2(\cdot)$ be the twice differentiable log densities of $Y$ and $N_X$ respectively. For compact notation, define*

$$\nu_{X|Y}(x|y) = \log\left(p_{X|Y}(x|y)\right)$$
$$= \log\left(p_{N_X}\left(\frac{x-h(y)}{k(y)}\right)/k(y)\right)$$
$$= \nu_2\left(\frac{x-h(y)}{k(y)}\right) - \log(k(y)) \quad and$$
$$G(x,y) = g(x) f'(x) + g'(x) [y - f(x)].$$

*Assume that $f(\cdot)$, $g(\cdot)$, $h(\cdot)$, and $k(\cdot)$ are twice differentiable. Then, the data generating mechanism must fulfill the following PDE for all $x, y$ with $G(x,y) \neq 0$.*

$$0 = \nu_1''(y) + \frac{g'(x)}{G(x,y)}\nu_1'(y) + \frac{\partial^2}{\partial y^2}\nu_{X|Y}(x|y) +$$
$$\frac{g(x)}{G(x,y)}\frac{\partial^2}{\partial y \partial x}\nu_{X|Y}(x|y) + \frac{g'(x)}{G(x,y)}\frac{\partial}{\partial y}\nu_{X|Y}(x|y).$$

To extend the above result to the multivariate setting, Strobl & Lasko (2023) introduce the restricted heteroscedastic noise model (HNM). For $j = 1, \ldots, d$, let $\mathrm{ND}_{Y^{(j)}}^{\mathcal{G}}$ denote the set of non-descendants of $Y^{(j)}$ in DAG $\mathcal{G}$. Then, following Definition 3 in Strobl & Lasko (2023), the location-scale noise model defined in Definition 3.1 is a restricted HNM if for all $j = 1, \ldots, d$, $f^{(j)}$ is nonlinear, and if for all $X \in \mathrm{PA}_{Y^{(j)}}^{\mathcal{G}}$ and $S$ satisfying $\mathrm{PA}_{Y^{(j)}}^{\mathcal{G}} \setminus \{X\} \subseteq S \subseteq \mathrm{ND}_{Y^{(j)}}^{\mathcal{G}} \setminus \{X, Y^{(j)}\}$, there exists $S = s$ where $p(s) > 0$ and Theorem A.2 is violated with respect to $p(x, y|s)$. Theorem 3 in Strobl & Lasko (2023) then establishes that if the data is generated by a HNM, the underlying causal graph is uniquely identifiable.

# B. Computational Experiments

## B.1. Notes on Implementation and Experimental Setup

We performed all experimental runs of SKEWD on CPUs of type Intel Xeon E5-2640v4 and Intel Xeon E5-4640v4 utilizing less than 10 GB RAM. We used Python version 3.11.5 on a Linux operating system. In the following, we provide details on the Bayesian optimization performed within SKEWD. We then state the hyperparameter configurations used within SKEWD throughout all of the experiments analyzed in Section 6.

**Bayesian Optimization** To tune the penalty parameters, we employ a cross-validated model selection procedure based on Bayesian optimization (Frazier, 2018). We generate initial candidate values jointly for $\alpha$ and $\kappa$ using Latin Hypercube Sampling on the log-scale and perform a $k$-fold cross-validation for each candidate. We maximize the penalized log-likelihood on the $k-1$ training folds and evaluate the unpenalized log-likelihood on the held-out fold, since the goal is to identify penalty parameters that improve the generalization of the actual likelihood function. The average over the $k$ unpenalized held-out likelihoods is the quantity being optimized through the Bayesian optimization, where the penalty parameters with the highest log-likelihood will be selected. While other optimizers could be employed in this heuristic framework, we use the CMA-ES in its implementation by Nomura & Shibata (2024) due to its robustness w.r.t. complex search landscapes with multiple local optima and sharp ridges.

**Hyperparameter Configurations** In the bivariate experiments, we set $\widetilde{q} = q = 14$ and $\widetilde{p} = p = 7$, while in the multivariate experiments, we set $\widetilde{q} = 10$ and $\widetilde{p} = 7$ for the B-splines, assuming that the scale function $g$ is less complex than the location function $f$. In the CMA-ES optimization, we always used an adaptive learning rate, a step size of 1, and a population size of 100. The optimizer was set to terminate after a maximum of 5000 iterations or when the parameters barely changed within 25 iterations, specifically if $||\boldsymbol{\theta}^{(k+25)} - \boldsymbol{\theta}^{(k)}||_2 < 10^{-6}$. We also specified bounds for certain parameters for stability: $\boldsymbol{\rho} \in [-10, 5]^p$ and $\lambda \in [-25, 25]$. The restriction on $\lambda$ has a negligible effect on the skewness range, as the maximum

absolute skewness that can be modeled through this restriction equals $0.9887$, which is only slightly smaller than the actual maximum of $0.9953$. For the ECM algorithm, we allowed for a maximum of 3000 iterations (multivariate) and 5000 iterations (bivariate), except for Tübingen pair 69, where we limited the iterations to 500 due to the large size of the dataset consisting of $n = 16382$ observations.

In the Bayesian optimization, we optimized the penalized likelihood on $k = 8$ folds (bivariate) and $k = 5$ folds (multivariate) with respect to $\alpha, \kappa$. As starting values for the final heuristic estimation, we used the $k$ parameter estimates for $\theta$ which were obtained from the training folds during Bayesian optimization associated with the best pair $\alpha, \kappa$. Additionally, further random starting values could be used for more robustness with respect to initial values.

### B.2. Graphical Comparison of Independence: Normal and Skew-Normal Distribution

SKEWD-IT fits skew-normal LSNMs in both directions $X \rightarrow Y$ and $Y \rightarrow X$. Based on the residuals from each model, the causal direction is inferred by comparing the independence of the cause-noise pairs. The independence test version of LOCI works similarly, but their estimation method assumes normally distributed noise.

For both SKEWD and LOCI, Figure 3 shows fits and estimated residuals in the causal direction $X \rightarrow Y$ (3a, left) and the anti-causal direction $Y \rightarrow X$ (3b, right) for an exemplary pair from the novel skew-noise LSs(1.750) dataset. In both subplots of Figure 3, the skew-normal model is depicted right, while the normal model is depicted left. The upper rows display the estimated mean and confidence intervals, whereas the cause-residual pairs are depicted in the lower row. In the causal direction (a), the estimated residuals in the normal-model show a dependence with cause $X$, indicated by the pointy outer contour level of the kernel density estimate (KDE) for small $X$, which is not present in the skew-normal model. In the normal-model, this dependence is incorrectly assessed as stronger by the HSIC test than the dependence exhibited in the anti-causal direction, seen in Figure 3b. The skew-normal model, which is a generalization of the normal model, overcomes the shortcoming on this data pair by modeling the skewness, leading to seemingly independent residuals (oval KDE contour levels) in the causal direction compared to the anti-causal direction. Ultimately, LOCI infers the wrong direction, whereas SKEWD infers the true causal direction.

*Figure 3.* Comparison of LSNM mean fits, confidence intervals and residuals based on the normal distribution via LOCI (left in both (a) and (b)) and skew-normal distribution via SKEWD (right in both (a) and (b)) for pair 6 from the novel LSs(1.750) dataset.

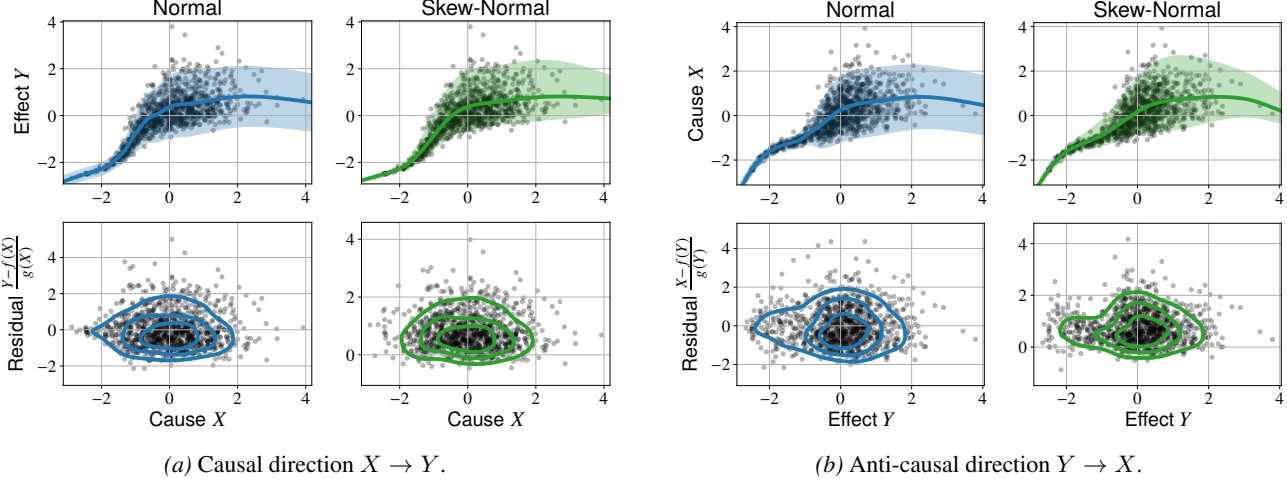

(a) Causal direction $X \rightarrow Y$.             (b) Anti-causal direction $Y \rightarrow X$.

### B.3. Ablation of the ECM Algorithm

In this section, we investigate the performance improvements achieved by the ECM algorithm compared to the initial heuristic solution based on the bivariate experiments discussed in Section 6. In Table 3, we show the accuracies on the novel skew-noise datasets for SKEWD-LL and SKEWD-IT as well as the accuracies that would have been achieved, if the heuristic parameter estimates via the CMA-ES in both directions were used to infer the causal directions. We denote the corresponding cause-effect methods as CMA-ES-LL and CMA-ES-LL. In the ANs case, the performance of the heuristic solutions is perfect in terms of likelihood scoring. However, when using independence testing, the refinement via ECM-algorithm in

SKEWD leads to an increase in accuracy of $5\%$ and $8.5\%$ on the ANs(-0.455) and ANs(0.985) dataset, respectively. On the ANs(1.750) dataset, both independence test versions achieve perfect accuracy.

The improvements in the performance on the causal task through the ECM algorithm in the LSs case are more critical. In that case, also the performance when using likelihood scoring is improved upon. Most notably by $12.5\%$ on the LSs(0.985) dataset. The improvements for independence-based inference are much larger, ranging between $28\%$ and $34\%$, demonstrating the necessity of refining the heuristic solution. This is further implied by the results in terms of AUDRC depicted in Table 4. Since they are mostly the same as in terms of accuracy, we do not further discuss them.

*Table 3.* Accuracy in % over skew-noise datasets for SKEWD compared to the results if log-likelihood scoring and independence testing was performed based on the output of the CMA-ES-based heuristic. We denote these two versions of the heuristic as CMA-ES-LL and CMA-ES-IT.

| Method | ANs(-0.455) | ANs(0.985) | ANs(1.750) | LSs(-0.455) | LSs(0.985) | LSs(1.750) |
|---|---|---|---|---|---|---|
| SKEWD-LL | 100.0 | 100.0 | 100 | 100.0 | 100.0 | 100 |
| CMA-ES-LL | 100.0 | 100.0 | 100 | 99.5 | 87.5 | 100 |
| SKEWD-IT | 98.5 | 97.5 | 100 | 86.0 | 82.5 | 92 |
| CMA-ES-IT | 93.5 | 89.0 | 100 | 52.0 | 54.5 | 64 |

*Table 4.* AUDRC in % over skew-noise datasets for SKEWD compared to the results if log-likelihood scoring and independence testing was performed based on the output of the CMA-ES-based heuristic. We denote these two versions of the heuristic as CMA-ES-LL and CMA-ES-IT.

| Method | ANs(-0.455) | ANs(0.985) | ANs(1.750) | LSs(-0.455) | LSs(0.985) | LSs(1.750) |
|---|---|---|---|---|---|---|
| SKEWD-LL | 100.00 | 100.00 | 100 | 100.00 | 100.00 | 100.00 |
| CMA-ES-LL | 100.00 | 100.00 | 100 | 99.99 | 93.68 | 100.00 |
| SKEWD-IT | 99.96 | 99.30 | 100 | 96.70 | 78.72 | 96.70 |
| CMA-ES-IT | 86.08 | 82.41 | 100 | 26.18 | 28.22 | 42.09 |

## B.4. Tabulated Results from Bivariate Experiments

In this section, we present the exact results achieved by all baselines on the novel skew-noise datasets used in the bivariate experiments from Section 6. Table 5 reports the accuracy (in %) and Table 6 shows the AUDRC (in %) on the skew-noise datasets.

The results with respect to the established benchmarks are presented in Table 7 showing the accuracy (in %) and Table 8 showing the AUDRC (in %). Moreover, we report the mean accuracy and AUDRC (both in %) reached by all baselines across all 19 considered datasets in Table 9.

*Table 5.* Accuracy in % over skew-noise datasets.

| Method | ANs(-0.455) | ANs(0.985) | ANs(1.750) | LSs(-0.455) | LSs(0.985) | LSs(1.750) |
|---|---|---|---|---|---|---|
| SKEWD-LL | 100.0 | 100.0 | 100 | 100.0 | 100.0 | 100 |
| SKEWD-IT | 98.5 | 97.5 | 100 | 86.0 | 82.5 | 92 |
| LOCI-LL | 100.0 | 99.5 | 83 | 100.0 | 96.0 | 66 |
| LOCI-IT | 99.5 | 99.5 | 98 | 72.0 | 69.0 | 75 |
| ROCHE | 99.5 | 97.5 | 95 | 99.5 | 88.5 | 81 |
| QCCD | 56.0 | 43.5 | 41 | 95.5 | 21.5 | 11 |
| GRCI | 87.5 | 80.5 | 93 | 79.5 | 69.5 | 84 |
| CAM | 100.0 | 100.0 | 100 | 82.5 | 70.0 | 33 |
| RESIT | 100.0 | 100.0 | 100 | 19.0 | 43.0 | 34 |
| IGCI | 38.0 | 38.5 | 56 | 30.5 | 53.5 | 88 |
| IGCI-G | 100.0 | 100.0 | 100 | 100.0 | 99.5 | 100 |

*Table 6.* AUDRC in % over skew-noise datasets.

| Method | ANs(-0.455) | ANs(0.985) | ANs(1.750) | LSs(-0.455) | LSs(0.985) | LSs(1.750) |
|--------|-------------|------------|------------|-------------|------------|------------|
| SKEWD-LL | 100.00 | 100.00 | 100.00 | 100.00 | 100.00 | 100.00 |
| SKEWD-IT | 99.96 | 99.30 | 100.00 | 96.70 | 78.72 | 96.70 |
| LOCI-LL | 100.00 | 100.00 | 96.06 | 100.00 | 99.69 | 85.34 |
| LOCI-IT | 99.99 | 99.99 | 99.89 | 87.28 | 56.98 | 91.39 |
| ROCHE | 100.00 | 99.96 | 99.85 | 100.00 | 99.28 | 97.97 |
| QCCD | 57.15 | 26.68 | 27.93 | 99.68 | 0.62 | 1.41 |
| GRCI | 82.86 | 75.00 | 98.87 | 87.43 | 67.03 | 96.33 |
| CAM | 100.00 | 100.00 | 100.00 | 74.08 | 59.61 | 10.24 |
| RESIT | 100.00 | 100.00 | 100.00 | 47.79 | 70.50 | 43.61 |
| IGCI | 45.93 | 30.81 | 71.32 | 45.74 | 56.56 | 96.83 |
| IGCI-G | 100.00 | 100.00 | 100.00 | 100.00 | 99.99 | 100.00 |

*Table 7.* Accuracy in % over established benchmark datasets. The Tübingen benchmark is abbreviated by Tue.

| Method | AN | ANs | LS | LSs | MNU | SIM | SIMc | SIMln | SIMG | Tue | Cha | Net | Multi |
|--------|----|-----|----|----|-----|-----|------|-------|------|-----|-----|-----|-------|
| SKEWD-LL | 100 | 100 | 100 | 100 | 87 | 54 | 51 | 81 | 88 | 45.97 | 54.33 | 73.33 | 72.00 |
| SKEWD-IT | 100 | 100 | 87 | 97 | 88 | 78 | 82 | 86 | 81 | 66.57 | 77.00 | 84.33 | 78.33 |
| LOCI-LL | 100 | 100 | 100 | 100 | 100 | 49 | 50 | 80 | 78 | 53.00 | 43.00 | 77.00 | 72.00 |
| LOCI-IT | 100 | 100 | 94 | 89 | 100 | 78 | 82 | 73 | 78 | 60.84 | 72.67 | 86.67 | 79.00 |
| ROCHE | 100 | 100 | 100 | 100 | 100 | 78 | 82 | 77 | 81 | 77.27 | 74.00 | 87.67 | 83.00 |
| QCCD | 100 | 82 | 100 | 96 | 99 | 62 | 72 | 80 | 64 | 77.09 | 53.67 | 80.33 | 50.67 |
| GRCI | 100 | 94 | 98 | 87 | 88 | 77 | 77 | 77 | 70 | 81.59 | 70.00 | 84.67 | 77.33 |
| CAM | 100 | 100 | 100 | 53 | 86 | 57 | 60 | 87 | 81 | 57.77 | 46.67 | 78.33 | 34.67 |
| RESIT | 100 | 100 | 60 | 3 | 5 | 78 | 82 | 87 | 77 | 62.03 | 34.33 | 78.33 | 37.33 |
| IGCI | 20 | 35 | 46 | 34 | 11 | 37 | 45 | 51 | 53 | 68.03 | 55.00 | 55.33 | 92.33 |
| IGCI-G | 100 | 100 | 98 | 99 | 100 | 38 | 39 | 59 | 83 | 56.34 | 58.00 | 59.67 | 68.00 |

*Table 8.* AUDRC in % over established benchmark datasets. The Tübingen benchmark is abbreviated by Tue.

| Method | AN | ANs | LS | LSs | MNU | SIM | SIMc | SIMln | SIMG | Tue | Cha | Net | Multi |
|--------|----|-----|----|----|-----|-----|------|-------|------|-----|-----|-----|-------|
| SKEWD-LL | 100.00 | 100.00 | 100.00 | 100.00 | 97.58 | 64.58 | 64.33 | 94.57 | 94.95 | 54.57 | 57.50 | 81.93 | 92.01 |
| SKEWD-IT | 100.00 | 100.00 | 93.71 | 98.62 | 97.78 | 90.36 | 93.83 | 96.78 | 93.67 | 69.71 | 77.47 | 96.30 | 83.21 |
| LOCI-LL | 100.00 | 100.00 | 100.00 | 100.00 | 100.00 | 59.94 | 63.74 | 94.73 | 88.95 | 67.02 | 47.25 | 86.17 | 92.66 |
| LOCI-IT | 100.00 | 100.00 | 99.41 | 97.42 | 100.00 | 88.74 | 93.16 | 85.97 | 93.24 | 59.62 | 71.33 | 96.97 | 77.76 |
| ROCHE | 100.00 | 100.00 | 100.00 | 100.00 | 100.00 | 86.61 | 93.85 | 88.96 | 92.43 | 75.46 | 78.42 | 97.62 | 94.40 |
| QCCD | 100.00 | 91.23 | 100.00 | 99.87 | 99.99 | 70.69 | 83.31 | 92.19 | 76.30 | 84.38 | 60.52 | 93.89 | 62.68 |
| GRCI | 100.00 | 99.62 | 99.67 | 95.14 | 97.18 | 89.62 | 92.38 | 92.09 | 87.69 | 73.23 | 70.81 | 95.97 | 73.60 |
| CAM | 100.00 | 100.00 | 100.00 | 36.29 | 76.45 | 67.76 | 68.82 | 87.20 | 88.24 | 69.37 | 41.05 | 84.88 | 40.02 |
| RESIT | 100.00 | 100.00 | 70.62 | 2.96 | 0.49 | 75.23 | 84.43 | 82.93 | 68.43 | 78.72 | 48.12 | 81.45 | 67.92 |
| IGCI | 17.83 | 35.45 | 59.54 | 48.98 | 1.27 | 34.49 | 40.86 | 51.34 | 62.88 | 73.81 | 58.39 | 61.23 | 98.75 |
| IGCI-G | 100.00 | 100.00 | 99.94 | 99.99 | 100.00 | 35.47 | 34.04 | 70.15 | 94.97 | 68.84 | 55.86 | 58.44 | 86.07 |

*Table 9.* Mean and standard deviation of accuracy, AUDRC over all 19 benchmarks. The top two approaches are bolded.

| Method | Accuracy (%) ↑ | AUDRC (%) ↑ |
|---|---|---|
| SKEWD-LL | 84.56 ± 19.95 | 89.58 ± 16.30 |
| SKEWD-IT | **87.46 ± 9.53** | **92.78 ± 8.92** |
| LOCI-LL | 81.39 ± 20.60 | 88.50 ± 16.51 |
| LOCI-IT | 84.54 ± 12.71 | 89.43 ± 13.63 |
| ROCHE | **89.52 ± 9.92** | **94.99 ± 7.56** |
| QCCD | 67.64 ± 26.44 | 69.92 ± 33.23 |
| GRCI | 82.93 ± 9.07 | 88.13 ± 11.05 |
| CAM | 75.10 ± 23.17 | 73.90 ± 26.44 |
| RESIT | 63.16 ± 33.39 | 69.64 ± 29.92 |
| IGCI | 47.75 ± 20.20 | 52.21 ± 23.97 |
| IGCI-G | 81.97 ± 23.20 | 84.41 ± 23.16 |

## B.5. Additional Metrics for the Multivariate Experiments

In this section, we report the performance of SKEWD-MV and the baseline methods on the multivariate experiments in terms of precision, recall, and F1-score. The results are summarized in Table 10. Note that the mean F1-scores for HOST, GraN-DAG++, and CAM are computed only over instances for which both precision and recall are strictly greater than zero. Such instances occur for all of these baselines, except for HOST when $d = 10$. Across all three metrics, SKEWD-MV achieves the best performance for both $d = 6$ and $d = 10$ and constitutes the only reliable method. When provided with the oracle Markov equivalence class, SKEWD-MV-O performs even better, achieving near-perfect scores across all measures.

*Table 10.* Mean precision, recall and F1-score of SKEWD-MV, SKEWD-MV-O and baselines on synthetic data from the multivariate experiments. For GraN-DAG++, 5 error-terminated runs were excluded.

| METHOD | $d = 6$ | | | $d = 10$ | | |
|---|---|---|---|---|---|---|
| | PRECISION ↑ | RECALL ↑ | F1-SCORE ↑ | PRECISION ↑ | RECALL ↑ | F1-SCORE ↑ |
| SKEWD-MV | 0.84 | 0.76 | 0.80 | 0.82 | 0.69 | 0.74 |
| SKEWD-MV-O | 0.99 | 0.99 | 0.99 | 1.00 | 1.00 | 1.00 |
| HOST | 0.20 | 0.23 | 0.22 | 0.25 | 0.46 | 0.32 |
| GRAN-DAG++ | 0.21 | 0.32 | 0.25 | 0.13 | 0.23 | 0.15 |
| CAM | 0.20 | 0.48 | 0.29 | 0.14 | 0.65 | 0.23 |

## B.6. Kernel Density Estimation for Marginal Distributions

As an alternative to the Gaussian marginal assumption in the likelihood scoring variants SKEWD-LL (bivariate) and SKEWD-MV (multivariate), we investigate kernel density estimation (KDE) for marginal density estimation. Specifically, we consider a Gaussian kernel with a cross-validated bandwidth. We report results across all datasets from Section 6, comparing Gaussian and KDE-based marginal estimation in the causal discovery task. Accuracy and AUDRC for the bivariate skew-noise datasets are shown in Tables 11 and 12, while results for the established benchmark datasets are reported in Tables 13 and 14.

Performance on the synthetic multivariate datasets, measured in terms of SID and SHD, is summarized in Table 15, which also includes SKEWD-MV when provided with the oracle Markov equivalence class (SKEWD-MV-O). In addition, Table 16 reports results in terms of precision, recall, and F1-score.

Overall, the Gaussian marginal assumption yields superior performance whenever the Gaussianity assumption is satisfied or nearly satisfied. This includes the skew-noise datasets, the SIMG dataset, and the regular AN(s) and LS(s) datasets. Similarly, under multiplicative uniform noise (MNU) as well as in the multivariate settings, where source nodes are uniformly distributed nodes and conditionals are skew-normally distributed, KDE-based marginal estimation consistently performs worse. In contrast, for the real-world Tübingen benchmark and for synthetic benchmarks with non-Gaussian marginals (SIM, SIMc, SIMln), KDE leads to notable gains in performance, with accuracy improvements of up to 22% (observed on SIM). These findings suggests that exploring advanced density estimation methods may further enhance likelihood-based variants.

The observed behavior can be explained by the role of the marginal density estimates in the likelihood-based causal scores. Focusing on the bivariate case, under a candidate causal direction $X \rightarrow Y$, the joint density is estimated as $\hat{p}_{X \rightarrow Y}(x, y) = \hat{p}(x)\hat{p}(y|x)$, where $\hat{p}(y|x)$ is modeled via the skew-normal likelihood. If the marginal $p(x)$ is incorrectly modeled as Gaussian although the true marginal deviates substantially from this parametric assumption, the marginal is misspecified and its estimated value should be generally underestimated. As a result, in the correct causal direction, the estimated joint likelihood can be systematically underestimated under misspecification. This can obscure the asymmetry between the two directions and, in some cases, even favor the incorrect one.

KDE, in contrast, provides a nonparametric estimate of the marginal densities and is therefore not tied to a specific distributional assumption. This makes it inherently more robust to deviations from Gaussianity, as it avoids disproportional reductions of the joint likelihood in the true causal direction caused by marginal misspecification. This explains the improved performance on the SIM, SIMc, SIMln and Tübingen benchmarks. However, this increased flexibility does not automatically lead to better causal discovery. KDE may introduce higher variance and does not exploit structural assumptions that can be beneficial when the parametric model is approximately correct. This is reflected, for instance, in the multivariate experiments, where KDE-based marginal estimation is outperformed by the parametric variant despite uniformly distributed (i.e. non-Gaussian) source nodes. Overall, KDE works better on non-Gaussian marginals, since it avoids the misspecification of marginal distributions, but its additional flexibility may reduce efficiency when the deviations from the Gaussian are negligible.

*Table 11.* Accuracy in % over skew-noise datasets for SKEWD-LL and SKEWD-LL when using KDE for marginal estimation.

| Method | ANs(-0.455) | ANs(0.985) | ANs(1.750) | LSs(-0.455) | LSs(0.985) | LSs(1.750) |
|---|---|---|---|---|---|---|
| SKEWD-LL | 100.0 | 100.0 | 100 | 100.0 | 100.0 | 100 |
| SKEWD-LL (KDE) | 92.0 | 84.0 | 96 | 79.0 | 82.5 | 84 |

*Table 12.* AUDRC in % over skew-noise datasets for SKEWD-LL and SKEWD-LL when using KDE for marginal estimation.

| Method | ANs(-0.455) | ANs(0.985) | ANs(1.750) | LSs(-0.455) | LSs(0.985) | LSs(1.750) |
|---|---|---|---|---|---|---|
| SKEWD-LL | 100.00 | 100.00 | 100.00 | 100.00 | 100.00 | 100.00 |
| SKEWD-LL (KDE) | 95.76 | 88.17 | 99.28 | 93.41 | 87.17 | 93.62 |

*Table 13.* Accuracy in % over established benchmark datasets for SKEWD-LL and SKEWD-LL when using KDE for marginal estimation. The Tübingen benchmark is abbreviated by Tue.

| Method | AN | ANs | LS | LSs | MNU | SIM | SIMc | SIMln | SIMG | Tue | Cha | Net | Multi |
|---|---|---|---|---|---|---|---|---|---|---|---|---|---|
| SKEWD-LL | 100 | 100 | 100 | 100 | 87 | 54 | 51 | 81 | 88 | 45.97 | 54.33 | 73.33 | 72.00 |
| SKEWD-LL (KDE) | 100 | 99 | 100 | 89 | 68 | 76 | 69 | 85 | 65 | 64.33 | 56.33 | 86.33 | 82.67 |

*Table 14.* AUDRC in % over established benchmark datasets for SKEWD-LL and SKEWD-LL when using KDE for marginal estimation. The Tübingen benchmark is abbreviated by Tue.

| Method | AN | ANs | LS | LSs | MNU | SIM | SIMc | SIMln | SIMG | Tue | Cha | Net | Multi |
|---|---|---|---|---|---|---|---|---|---|---|---|---|---|
| SKEWD-LL | 100.00 | 100.00 | 100.00 | 100.00 | 97.58 | 64.58 | 64.33 | 94.57 | 94.95 | 54.57 | 57.50 | 81.93 | 92.01 |
| SKEWD-LL (KDE) | 100.00 | 99.98 | 100.00 | 98.83 | 78.25 | 88.59 | 86.90 | 96.52 | 80.80 | 64.05 | 58.27 | 95.48 | 93.94 |

*Table 15.* SHD and SID on synthetic benchmarks for SKEWD-MV and SKEWD-MV-O when assuming Gaussian marginals (regular) and when using KDE for marginal estimation.

| METHOD | d = 6 | | d = 10 | |
|---|---|---|---|---|
| | SHD ↓ | SID ↓ | SHD ↓ | SID ↓ |
| SKEWD-MV | 2.27 | 4.35 | 4.72 | 14.97 |
| SKEWD-MV (KDE) | 2.87 | 5.89 | 5.58 | 17.38 |
| SKEWD-MV-O | 0.14 | 0.23 | 0.04 | 0.10 |
| SKEWD-MV-O (KDE) | 0.88 | 2.07 | 1.08 | 3.31 |

*Table 16.* Mean precision, recall and F1-score on synthetic benchmarks for SKEWD-MV and SKEWD-MV-O when assuming Gaussian marginals (regular) and when using KDE for marginal estimation.

| METHOD | d = 6 | | | d = 10 | | |
|---|---|---|---|---|---|---|
| | PRECISION ↑ | RECALL ↑ | F1-SCORE ↑ | PRECISION ↑ | RECALL ↑ | F1-SCORE ↑ |
| SKEWD-MV | 0.84 | 0.76 | 0.80 | 0.82 | 0.69 | 0.74 |
| SKEWD-MV (KDE) | 0.78 | 0.72 | 0.74 | 0.76 | 0.65 | 0.70 |
| SKEWD-MV-O | 0.99 | 0.99 | 0.99 | 1.00 | 1.00 | 1.00 |
| SKEWD-MV-O (KDE) | 0.92 | 0.92 | 0.92 | 0.94 | 0.94 | 0.94 |

## C. Bivariate Skew-Noise Datasets

In this section, we display scatter plots of 4 randomly chosen exemplary pairs per novel skew-noise dataset. We refer to Figure 4 for ANs(-0.455), to Figure 4 for ANs(0.985), to Figure 6 for ANs(1.750), to Figure 7 for LSs(-0.455), to Figure 8 for LSs(0.985), and to Figure 9 for LSs(1.750).

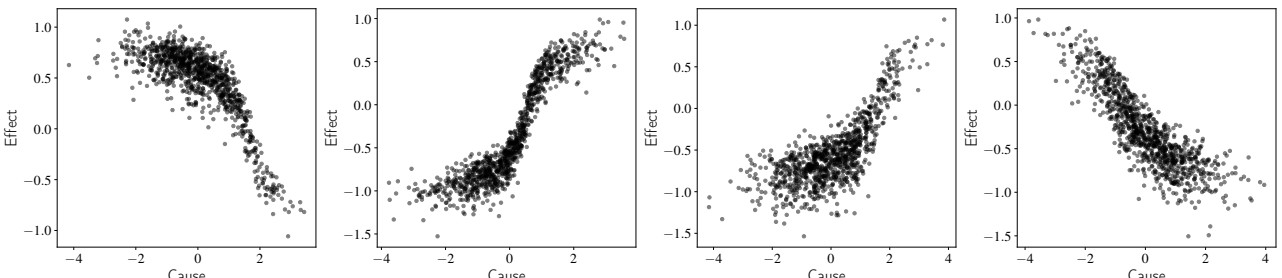

*Figure 4.* Exemplary pairs from the ANs(-0.455) dataset. Left two pairs generated with skew-normal noise, right two pairs generated with GNO noise.

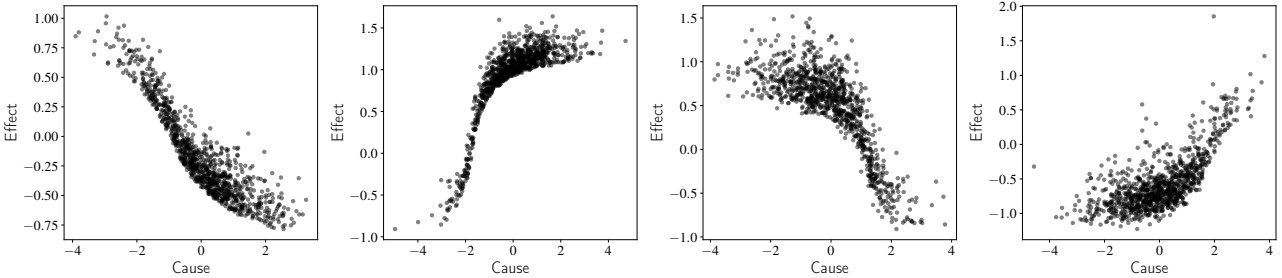

*Figure 5.* Exemplary pairs from the ANs(0.985) dataset. Left two pairs generated with skew-normal noise, right two pairs generated with GNO noise.

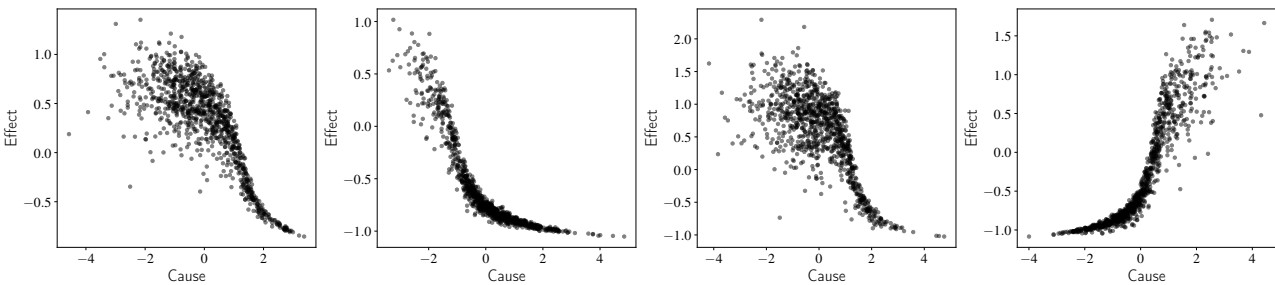

*Figure 6.* Exemplary pairs from the ANs(1.750) dataset.

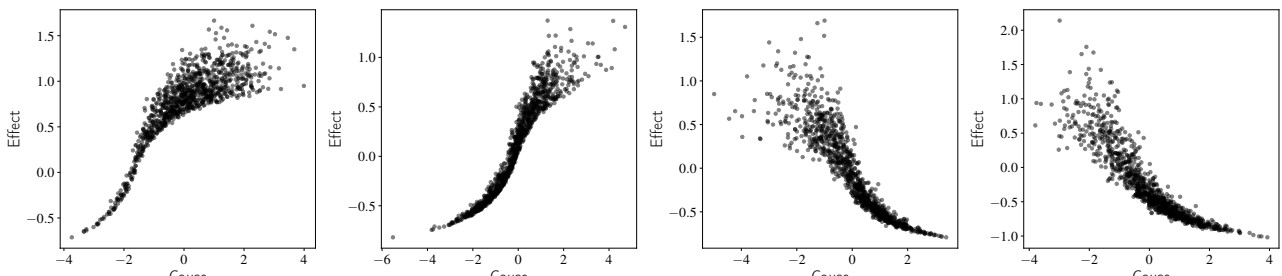

*Figure 7.* Exemplary pairs from the LSs(-0.455) dataset. Left two pairs generated with skew-normal noise, right two pairs generated with GNO noise.

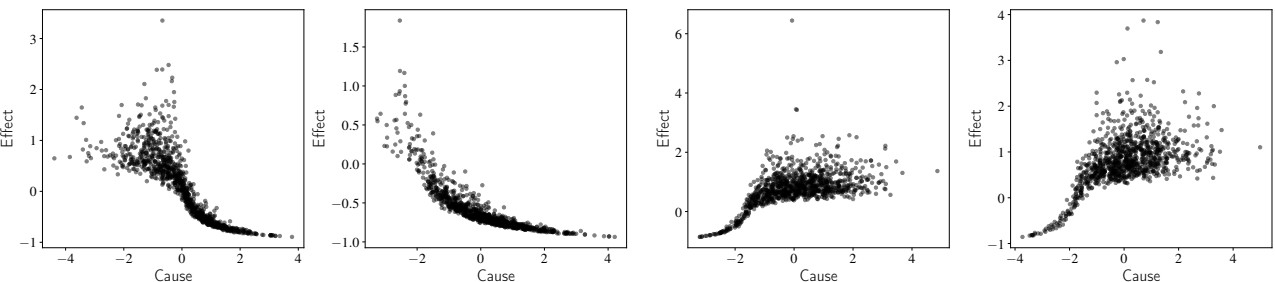

*Figure 8.* Exemplary pairs from the LSs(0.985) dataset. Left two pairs generated with skew-normal noise, right two pairs generated with GNO noise.

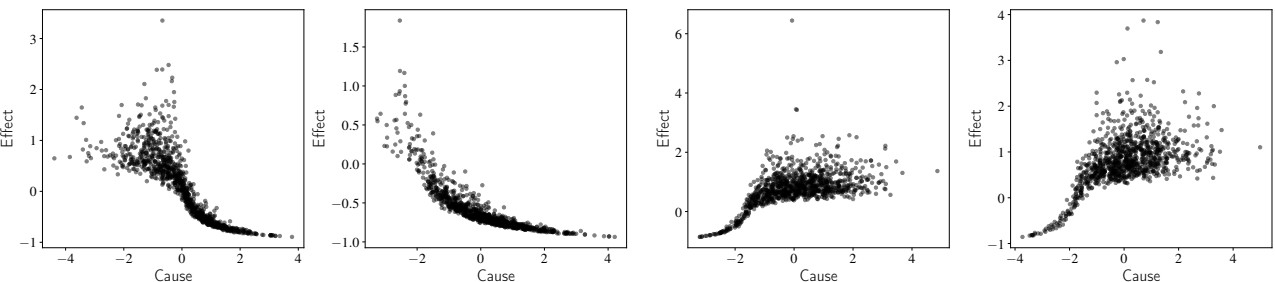

*Figure 9.* Exemplary pairs from the LSs(1.750) dataset.

