# OpenReview forum: "Skewness-Robust Causal Discovery in Location-Scale Noise Models"
_ICML.cc/2026/Conference — ICML 2026 regular_

### Official Review · Reviewer_eWCs · 2026-02-20

**Soundness:** 3
**Presentation:** 3
**Significance:** 3
**Originality:** 3
**Overall Recommendation:** 4
**Confidence:** 3

**Summary:**

This paper studies causal discovery for location-scale noise models (LSNMs), and provides a method that remains reliable when the noise distribution is skewed rather than symmetric. The authors focus on the bivariate cause–effect case under LSNMs, where most prior work assumes symmetric (often Gaussian) noise, and to extend their method to multivariate DAG learning. The proposed method, SKEWD, models the noise using a skew-normal distribution and performs causal direction inference. Two versions include residual independence testing (SKEWD-IT), and likelihood scoring (SKEWD-LL). Experiments are conducted to compare with many baselines.

**Compliance With Llm Reviewing Policy:**

Affirmed.

**Final Justification:**

The authors have addressed my concerns in the rebuttal. I keep my positive score unchanged.

**Key Questions For Authors:**

- What is the intuition that the existing methods assuming symmetry fail to recover the nosie variable? More discusion on this using Figure 1 for example is expected.
- The paper uses skewed normal to model the noise variable. How robust is this approach when the noise is subject to other asymmetric distributions? This is milidly mentioned In the experiment where GNO is examined. More discussion and comprehensive evaluations are expected.
- For causal discovery, authors propose a PC algorithm + likelihood scoring method, which requires faithfulness to operate. Why not simply adopt the RESIT framework as the identifiability follows from the similar idea?
- In the experiment evaluation, both in Figure 2 and table 1, many baselines give comparable and sometimes even better performances, e.g. ROCHE, LOCI, IGCI. Any comments on this observation? The advantage of the proposed method is not clear.
- In the experiment on causal discovery, since the dimension is not large, why only consider such a small set of baselines, but not the ones compared in the binary case?

**Limitations:**

The paper talks about limitaions on computational efficiency and misspefication of the skewed normal distributions. But more discussion is expected.

**Strengths And Weaknesses:**

**Stregnth:**
- The problem is clear and well-motivated. Most LSNM-based causal discovery methods assume symmetric noise. The authors argue that skewness is common in real data.
- Skew-normal distribution is a natural generalization to existing approaches. It nests the Gaussian case while only adds one additional parameter (shape).
- Comparehensive synthetic evaluation are conducted comparing with many baselines. Experiments show promising performance.

**Weakness:**
- The theoretical contribution is limited. The identifiability follows from prior LSNM theory. No analysis of consistency or statistical rates.
- Empirical evaluation could be improved. See questions.

---

> ### Author Rebuttal · Authors · 2026-03-30
>
> We thank the reviewer for their constructive feedback and valuable questions. Based on our answers below, we will revise the manuscript to expand the discussion on why current methods fail under skewness and to discuss the method's robustness with respect to skewed but non-skew-normal noise in more detail.
>
> **Why do methods assuming symmetry fail under skewness?**
>
> Skewness induces misspecification in likelihood models that assume symmetric noise, which in turn complicates the estimation of $f$ and $g$. Even small distortions in the estimated $f$ and $g$ can propagate to the residuals hindering independence between the cause and residual. This effect is illustrated in Fig. 1, where under a Gaussian assumption, the extracted residuals exhibit clear dependence, whereas the skew-normal model yields approximately independent residuals, even though the true skewness (1.75) in the GNO-noise exceeds the range of the skew-normal. Notably, the dependence in the Gaussian case appears to stem from an inadequate variance fit for small $x$ resulting in the pointy contour levels.
>
> In likelihood-based causal discovery, such misspecification leads to a systematic underestimation of the likelihood in the true causal direction, which can in turn favor the incorrect direction. This effect is evidenced in Fig. 2, where the performance of LOCI-LL deteriorates steadily with increasing skewness in both ANMs and LSNMs.
>
> **Robustness w.r.t. other asymmetric distributions**
>
> Our skew-LSNM benchmarks include three different parameterizations of the GNO distribution in addition to the skew-normal (SN) parametrizations. The GNO distribution constitutes a flexible family that generalizes both the normal and log-normal distribution (Hosking, 1986), allowing for strong deviations from the SN assumption.
>
> Specifically, we consider the GNO(0,1,$k$) with $k=0.15,-0.31,-0.5$. For $k=-0.5$, the skewness reaches 1.75, exceeding the maximum skewness of the SN, see Def. 3. Moreover, the GNO exhibits different tail behavior than the SN, having a truncated tail and potentially heavier tails. For $k=-0.31$ and $k=-0.5$, the excess kurtosis is 1.77 and 5.90, with the latter comparable to a more heavy-tailed exponential or t-distr. with $\nu=5$ degrees of freedom, while the SN can only exhibit a maximum excess kurtosis of 0.87.
>
> Despite these substantial deviations in both skewness and tail behavior, SkewD-LL achieves perfect accuracy across all GNO settings (Fig. 2). This demonstrates that our method is robust to misspecification well outside the SN family.
>
> **On using the RESIT framework**
>
> The RESIT framework heavily relies on a multivariate independence test to test the dependence between the noise and all causes which is a big issue. In comparison, SkewD-MV only requires bivariate (conditional) independence tests within the PC algorithm.
>
> **On the advantage of SkewD**
>
> SkewD-LL is the only method that achieves perfect accuracy across all skew-noise datasets. While IGCI-G performs similarly, it relies on a strong (and here violated) assumption of near deterministic functions and underperforms on the standard benchmark datasets. Further, it is strongly susceptible to the data preprocessing (normalization vs. standardization).
>
> On the challenging skew LSNM datasets, Figure 2 shows a clear degradation in performance for competing methods (ROCHE, LOCI, QCCD) as skewness increases. Even the best competitor (ROCHE) reaches only 0.81 accuracy at the highest skewness level, whereas SkewD-LL consistently maintains perfect accuracy, even when the skew-normal assumption is violated. This highlights the robustness of our approach and suggests that, under skewed-noise LSNMs, SkewD-LL is the method of choice. Another advantage of SkewD-LL over the competing LSNM approaches LOCI, ROCHE and QCCD is that it can be applied to the multivariate setting, where it also achieves strong performance on small-size graphs.
>
> On the established benchmark datasets, where the parametric assumptions are often violated, Table 1 shows that SkewD-IT outperforms SkewD-LL. This is consistent with findings by Sun and Schulte (2023) that independence-based methods are more robust under misspecification. In this setting, SkewD-IT ranks second overall, with only a $2.67$ percent gap in mean accuracy to ROCHE, and thus remains a highly competitive and reliable choice even when model assumptions are not fully met.
>
> **On the choice of multivariate baselines**
>
> In the multivariate experiments, we concentrated on currently dominating causal discovery methods. For instance, we excluded NOTEARS, since GraNDAG has been shown to be superior (Lachapelle et al., 2020). We did not include the LSNM methods compared in the bivariate setting, as these are limited to bivariate cause-effect inference. Instead, we consider the state-of-the-art methods HOST and GraNDAG++ for multivariate LSNMs and included CAM as an ANM baseline.

---

> > ### Author Rebuttal · Reviewer_eWCs · 2026-04-03
> >
> > I thank the authors for their response. My concerns are addressed. I keep my score unchanged.

---

### Official Review · Reviewer_KiKW · 2026-02-24

**Soundness:** 2
**Presentation:** 3
**Significance:** 2
**Originality:** 2
**Overall Recommendation:** 4
**Confidence:** 3

**Summary:**

This paper aims to enhance causal discovery within the framework of Location-Scale Noise Models (LSNMs) by relaxing the restrictive assumption of symmetric error distributions. This paper considers the setting where observational data exhibits significant skewness—a common feature in real-world domains like environmental science—which causes standard methods (such as LOCI) to fail due to model misspecification.

To address this, the paper introduces SKEWD (Skewness-Robust Discovery), a method that models the noise term using the Skew-Normal distribution, effectively generalizing the Gaussian assumption to handle asymmetric data. The framework employs Generalized Additive Models (GAMs) with B-splines to flexibly estimate non-linear location and scale functions. Recognizing the challenges of maximum likelihood estimation in this setting (e.g., non-convexity), the authors develop a robust hybrid optimization strategy that combines Bayesian optimization for hyperparameter selection, a heuristic search via CMA-ES, and a refinement step using an Expectation Conditional Maximization (ECM) algorithm.

For causal inference, SKEWD determines the causal direction either through residual independence testing (using HSIC) or by comparing penalized log-likelihood scores. The method is further extended to the multivariate setting (SKEWD-MV) to score Directed Acyclic Graphs (DAGs). Empirical evaluations demonstrate that SKEWD significantly outperforms state-of-the-art Gaussian-based approaches on skewed data benchmarks while maintaining competitive performance on symmetric data.

**Compliance With Llm Reviewing Policy:**

Affirmed.

**Final Justification:**

The authors have adequately addressed my main concerns in the rebuttal. I have increased my score accordingly

**Key Questions For Authors:**

1. Robustness to Model Misspecification Beyond Skew-Normal: This paper concentrates on the "skewed" data, yet the proposed method strictly assumes the noise follows a Skew-Normal distribution. In real-world scenarios, data may be asymmetric but follow different distributions (e.g., Log-Normal, Gumbel, or heavy-tailed asymmetric distributions). How does SKEWD perform when the underlying noise is skewed but not Skew-Normal?

1.1 Impact on Evaluation: If the method is highly sensitive to the specific parametric form of the skewness, the claim of being "robust" is overstated. Evidence of performance on non-Skew-Normal asymmetric noise would clarify the method's practical utility versus more general non-parametric approaches.

2. Optimization Complexity and Stability: I understand the authors want to provide a reliable discovery method, yet the optimization pipeline is exceptionally complex, requiring a cascade of Bayesian Optimization, CMA-ES, and ECM. This suggests the objective function is ill-behaved or riddled with local optima. Can you provide an ablation study or analysis showing the necessity of this three-stage pipeline? Specifically, how often does the heuristic search (CMA-ES) fail to find a basin of attraction that the ECM can refine?

2.1 Impact on Evaluation: If the method relies heavily on this specific, computationally expensive stack to function, it raises concerns about reproducibility and stability on new datasets. A clarification here would determine if the method is a robust tool or a fragile assembly of heuristics.


3. Comparison with Non-Parametric Baselines: Why was the method primarily compared against Gaussian-based parametric methods (like LOCI) rather than flexible non-parametric independence tests (e.g., HSIC-based methods without restrictive noise assumptions) or flow-based LSNMs?

3.1: Impact on Evaluation: If SKEWD does not significantly outperform a standard non-parametric method that makes no distributional assumptions, the value of introducing the complex Skew-Normal machinery is questionable. Comparing against a broader set of baselines would justify the need for this specific parametric constraint.

**Limitations:**

No. While the authors touch upon the challenges of optimizing the Skew-Normal likelihood, the discussion regarding the method's limitations and broader impact could be significantly strengthened. Specifically, I suggest the authors address the following points:

Parametric rigidity: The method is termed "robust," but it strictly assumes a Skew-Normal distribution. The authors should discuss the limitations of this assumption when dealing with real-world data that is asymmetric but follows a different distribution (e.g., Log-Normal, Gumbel, or heavy-tailed distributions), as this constitutes a different form of model misspecification.

Scalability: The proposed optimization pipeline (Bayesian Optimization + CMA-ES + ECM) appears computationally intensive. A discussion on the runtime complexity and the practical upper bound on the number of variables ($d$) for the multivariate extension would be valuable.

Societal Impact: A brief acknowledgement of the risks of applying causal discovery algorithms in high-stakes domains (e.g., healthcare, policy) would be appropriate, particularly noting that incorrect causal conclusions derived from misspecified models can lead to harmful interventions.

**Strengths And Weaknesses:**

1. Strengths

1.1. Relevance of Problem Setting: The paper correctly identifies that the assumption of Gaussian noise in Location-Scale Noise Models (LSNMs) is often violated in practice. Addressing skewness in causal discovery is a valid motivation.

1.2. Flexible Functional Form: The use of Generalized Additive Models (GAMs) with B-splines allows for non-linear relationships in both the location and scale functions, offering more flexibility than strictly linear models.

1.3 Detailed Optimization Scheme: The authors are transparent about the difficulties of fitting Skew-Normal distributions and provide a comprehensive (albeit complex) three-stage optimization strategy involving Bayesian Optimization, CMA-ES, and ECM.
Weaknesses

2. Weaknesses:

2.1. Limited Originality and Incremental Contribution: The core contribution is essentially substituting the Gaussian noise assumption in LSNMs with a Skew-Normal distribution. While this paper concentrates on skewed data, the methodological novelty is limited. This feels like a direct "plug-and-play" extension of existing frameworks (like LOCI) rather than a fundamental advancement in causal discovery theory. The reliance on known properties of the Skew-Normal distribution and standard GAM techniques results in a method that is an incremental engineering improvement rather than a significant conceptual leap.

2.2 Soundness and Optimization Complexity: A major concern regarding the technical soundness is the extreme complexity required to estimate the model parameters. The proposed solution necessitates a cumbersome three-step pipeline: (1) Bayesian Optimization for hyperparameters, (2) CMA-ES for initialization, and (3) an ECM algorithm for refinement. This heavy reliance on such an elaborate heuristic stack suggests that the objective function is ill-behaved and difficult to optimize reliably. This raises significant doubts about the method's robustness and stability on diverse real-world datasets where "ground truth" hyperparameters are unknown. If a method requires this level of tuning and hybrid optimization to converge, its practical utility is severely diminished compared to more stable, albeit simpler, baselines.

2.3. Narrow Scope of "Robustness": This paper aims to provide "skewness-robust" discovery, but the solution is parametrically rigid. By committing specifically to the Skew-Normal distribution, the method may fail when data exhibits other common non-Gaussian characteristics, such as heavy tails (Student-t) or multimodality, which are not captured by the Skew-Normal shape parameter. A truly "robust" method would likely employ a non-parametric approach or a more general family of distributions. Consequently, the claim of robustness is overstated; the method is arguably only "Skew-Normal robust," not generally robust to asymmetric noise.

2.4. Significance and Scalability: The computational cost of the proposed optimization pipeline is likely prohibitive for higher-dimensional problems. While the authors present a multivariate extension, the reliance on the PC algorithm to first find the skeleton, followed by this expensive pairwise scoring, restricts the method's applicability to small-scale graphs. Given the existence of non-parametric independence tests that do not require explicit density estimation (and thus avoid the optimization headaches described here), the practical advantage of SKEWD is not convincingly argued.

---

> ### Author Rebuttal · Authors · 2026-03-30
>
> We thank the reviewer for their valuable feedback and will revise the manuscript to address the highlighted limitations discussed below. In particular, we will
>
> - expand the discussion on robustness under misspecification,
> - stress advantages of SkewD over nonparametric baselines,
> - clarify multivariate scalability,
> - include a societal impact statement, acknowledging risks in high-stakes domains.
>
> Answers to key questions:
>
> **1. Robustness w.r.t. other asymmetric distributions**
>
> We agree that robustness beyond the skew-normal (SN) assumption requires further discussion. Our skew-LSNM benchmarks include three parameterizations of the GNO, which generalizes both the normal and log-normal distribution (Hosking, 1986), allowing for strong deviations from the SN.
>
> We consider the GNO(0,1,$k$) with $k=0.15,-0.31,-0.5$. For $k=-0.5$, the skewness reaches 1.75, exceeding the SN range ~(-1,1) (Def. 3). The GNO further differs in tail behavior, having a truncated tail and potentially heavier tails. For $k=-0.31,k=-0.5$, the excess kurtosis is 1.77 and 5.90. The latter is comparable to a more heavy-tailed exponential or t-distr. with $\nu=5$, while the maximum value of the SN is 0.87.
>
> Despite these substantial deviations, SkewD-LL achieves perfect accuracy across all GNO settings (Fig. 2), demonstrating robustness well beyond the SN family.
>
> More flexible alternatives (e.g., fully nonparametric) increase expressiveness but may come at the cost of more difficult optimization and reduced performance (see answer 3).
>
> **2. Ablation of the optimization pipeline**
>
> An ablation study is provided in Appendix B.3 in the manuscript (referenced in Section 4), demonstrating the necessity of the refinement via the ECM algorithm for reliable causal discovery in LSNMs when using our method, especially when resorting to SkewD-IT. On all skewed LSNM datasets, the accuracy improvements on SkewD-IT gained by the ECM refinement range between 0.28 and 0.34. For instance, for the LSs(0.985) dataset, the accuracy increases from 0.545 to 0.825. In contrast, for SkewD-LL, the heuristic alone is sufficient on ANM data, but beneficial for LSNMs, improving accuracy from 0.875 to 1 on the LSs(0.985) dataset.
>
> W.r.t. to a potential instability on new benchmarks, we believe that our proposed pipeline improves robustness rather than adding unnecessary complexity, given the extensive evaluation across 13 established benchmark datasets (in addition to our newly proposed datasets). While the necessity of the ECM step becomes clear from the previous paragraph, we argue for the Bayesian optimization that learning hyperparameters via optimization avoids reliance on fixed, potentially suboptimal choices and allows the method to adapt to varying data characteristics.
>
> **3. Comparison with nonparametric baselines**
>
> While many considered LSNM baselines (e.g., LOCI, ROCHE) are parametric, we did include nonparametric approaches. In particular, QCCD (based on nonparam. quantile regression) clearly degrades in accuracy with increasing skewness (Fig. 2).
>
> We also evaluated CAREFL (normalizing flows), which also underperforms compared to SkewD. It was not included as a baseline, since we were unable to reproduce the reported results on the Tübingen dataset, obtaining much lower performance. For reference, CAREFL achieves decreasing accuracy on the skew-LSNM datasets with increasing skewness (0.895, 0.71, 0.66), and achieves accuracies $\leq 0.44$ on all SIM datasets. We will add a discussion and provide both code and experimental results.
>
> Overall, these results indicate that SkewD outperforms both parametric and nonparametric state-of-the-art baselines under skewed LSNMs.
>
> **Scalability**
>
> The computational cost of SkewD-MV is mainly driven by the need to fit a skew-normal model for each admissible parent-child configuration (pcc) arising from undirected edges in the Markov equivalence class (MEC). Disregarding the DAG property, the number of such configurations for a specific parent node is $2^k-1$ given it has $k$ undirected edges, which limits scalability. Moreover, the higher the number of children in a pcc, the more parameters need to be estimated.
>
> Thus, scalability depends rather on the MEC than on the number of variables $d$. If the MEC contains relatively few admissible pccs, the number of configurations remains manageable even in larger graphs. For example, in our experiments with $d=10$, the maximum number of configurations was 40, so that the order of 100 pccs should still be handled by SkewD.
>
> Accordingly, we view the method as most suitable for small-scale graphs, where it provides reliable inference. It was primarily developed for the bivariate setting, while the multivariate extension should be understood as a proof of concept. Given the sufficiency of the heuristic in the bivariate case, a practical way to improve scalability is to omit the ECM step, which may reduce runtime at the cost of performance.

---

> > ### Author Rebuttal · Reviewer_KiKW · 2026-03-31
> >
> > The authors have adequately addressed my main concerns in the rebuttal. I have increased my score accordingly

---

### Official Review · Reviewer_SiZh · 2026-03-11

**Soundness:** 4
**Presentation:** 3
**Significance:** 3
**Originality:** 3
**Overall Recommendation:** 5
**Confidence:** 4

**Summary:**

This paper focuses on determining a unique DAG from a Markov equivalence class under location-scale noise models (LSNM), which generalizes additive noise models. More specifically, it focuses on determining the causal direction between two random variables $X$ and $Y$. The paper relaxes the symmetric distribution assumptions, made by most existing work, on the noise variable in LSNMs. The paper proposes a causal discovery algorithm named SKEWD for LSNMs with a skewed normal noise distribution. The paper primarily focuses on bivariate causal-effect inference, but it also shows how to extend the proposed method to the multivariate setting. Empirical experiments show that SKEWD outperforms SOTA baselines for LSNMs with skew-normal noise.

**Compliance With Llm Reviewing Policy:**

Affirmed.

**Final Justification:**

The authors have addressed my concerns. I have also read the rebuttal to other reviews. I do not see major concerns that will change my original assessment, given the technical contribution of the paper over the state-of-the-art baselines. The original manuscript is well-written. Incorporating the suggestions from other reviewers will make it even better.

**Key Questions For Authors:**

1. Have the authors looked into how much more efficient the heuristic optimization step is?
2. Can the authors provide any intuition about why using kernel density estimation for non-Gaussian marginals seems to work reasonably well?
3. What suggestions would the author have if one were to use this for large-scale causal discovery? How scalable will this method be?

**Limitations:**

Yes

**Strengths And Weaknesses:**

**Soundness **
- The derivations of E and CM-steps from lines 214 to the second column lines 227 are sound and rigorous.
- Proposition 5.1 is correct.

**Presentation**
- The paper is very well-written with clear explanations and mathematical derivations.
- The paper gives many technical details in the appendix with clarity for readers to understand with clarity.
- Figure 1 is good for readers to quickly understand the contributions of the paper.
- SKEWD-IT should be defined before using it in the limitations section under section 5.3.

**Significance**
- Line 202: it is a bit confusing when the paper says to assume fixed values for $\alpha$ and $\kappa$ and then use Bayesian optimization to approximate them.
- The paper adopts a heuristic to optimize the penalized log-likelihood using the covariance matrix adaption evolution strategy by (Hansen & Ostermeier, 2001; Hansen, 2016) to reduce the computational costs for maximizing the penalized log-likelihood. Still, using multiple starting values for $\theta$ can also be inaccurate. To mitigate this issue, the paper uses Bayesian optimization to learn what $\alpha$ and $\kappa$ should be, it seems like the overall efficiency is still better than using an ECM algorithm directly. This increases both efficiency while maintaining desirable accuracy in many cases as shown by Table 3 and Table 4 for the likelihood scoring variant CMA-ES-LL.
- The paper proposes a causal discovery algorithm to distinguish Markov equivalent DAGs in LNSMs with a skewed normal noise distribution, which is new and more practical. The experimental section supports the theoretical advantages of the proposed method in highly skewed noise cases for cause-effect inference.
- The paper also provides pseudocode to extend the proposed method to multivariate settings.
- The results on nonparametric estimation of the marginal distributions via kernel density estimation seems promising based on Tables 5, 6, 7, even when there may be nonidentifiability.
- The method outperforms reasonable baselines in causal discovery.

**Originality**
- The paper draws on different existing work e.g. ECM algorithm (Ferreira et al. (2023)), covariance matrix adaptation evolution strategy (Hansen & Ostermeier, 2001; Hansen, 2016), rewriting skew-normal distribution (Ferreira & Paula (2017)), to optimize for the penalized log-likelihood for causal discovery under LSNMs with skew-normal noise distribution.
- The paper employs multiple modifications in order to optimize the penalized log-likelihood: 1. on top of using the ECM algorithm for partially nonlinear models, the paper drops the additional linear regression term and model the scale parameter through splines rather than linear regression and introduce a penalty $\kappa$ for the scale and extend the algorithm to the GAM setting 2. using Bayesian optimization to select $\alpha$ and $kappa$.

---

> ### Author Rebuttal · Authors · 2026-03-30
>
> We thank the reviewer for the positive assessment of our manuscript and for the thoughtful questions. We provide detailed responses to the questions below and we will update our manuscript with according clarifications.
>
> **1. How much more efficient is the heuristic optimization?**
>
> We indeed have looked into how effective the estimation of the heuristic optimization is with respect to the causal discovery task. The ablation study can be found in Appendix B.3. There, we discuss the performances on the causal discovery task obtained on our synthetic benchmarks when using the combination of heuristic and ECM vs. when using only the heuristic for optimization.
>
> When using likelihood scoring, the heuristic is relatively sufficient when the data is generated by an ANM (perfect accuracy), whereas on the LSs(0.985) dataset, the ECM is beneficial, improving accuracy of 0.875 to 1. When one wants to resort to the independence-based approach, which has been shown to be more robust w.r.t. various distribution settings, the ECM algorithm becomes more critical. While the improvements are rather small on ANMs, on LSNMs, the accuracy improvements gained by the ECM refinement range between 0.28 and 0.34 for all three considered datasets. For instance, for the LSs(0.985) dataset, the accuracy increases from 0.545 to 0.825
>
> **2. Why does using kernel density estimation for non-Gaussian marginals seem to work reasonably well?**
>
> The key intuition is that kernel density estimation (KDE) avoids a source of misspecification that can distort likelihood-based causal scoring. In our likelihood-based approach, the joint density under candidate causal direction $X \rightarrow Y$, i.e. $X$ causes $Y$, is estimated as $\hat{p}_{X \rightarrow Y}(x,y) = \hat{p}(x) \hat{p}(y|x)$, where $\hat{p}(y|x)$ corresponds to the estimated skew-normal likelihood. If ${p}(x)$ is modeled as Gaussian although the true marginal deviates substantially from this parametric assumption, the marginal is misspecified and its estimated value should be generally small. As a result, in the correct causal direction, the estimated joint likelihood can be systematically underestimated under misspecification. This can obscure the asymmetry between the two directions and, in some cases, even favor the incorrect one.
>
> KDE, in contrast, provides a nonparametric estimate of the marginal densities and is therefore not tied to a specific distributional assumption. This makes it inherently more robust to deviations from Gaussianity, as it avoids disproportional reductions of the joint likelihood in the true causal direction caused by marginal misspecification. It is therefore not surprising that KDE leads to better results w.r.t. the SIM, SIMc, SIMln and Tübingen benchmarks (Appendix B.4.).
>
> However, this increased flexibility does not automatically lead to better causal discovery. KDE may introduce higher variance and does not exploit structural assumptions that can be beneficial when the parametric model is approximately correct. This is reflected, for instance, in our multivariate experiments: even though source nodes are generated from non-Gaussian (uniform) distributions, SkewD-MV with KDE performs worse than the parametric SkewD-MV.
>
> Overall, KDE works better on non-Gaussian marginals, since it avoids the misspecification of marginal distributions, but its additional flexibility may reduce efficiency when the deviations from the Gaussian are negligible.
>
> **3. What suggestions would the author have if one were to use this for large-scale causal discovery? How scalable will this method be?**
>
> Our method is not designed for large-scale causal discovery. Its computational cost is driven by the need to fit a skew-normal model for each admissible parent-child configuration (pcc) arising from undirected edges in the Markov equivalence class (MEC). Disregarding the DAG property, the number of such configurations for a specific parent node is $2^k-1$ given it has $k$ undirected edges, which limits scalability.
>
> Accordingly, we view the method as most suitable for smaller-scale problems, where it provides reliable inference. It was primarily developed for the bivariate setting, while the multivariate extension should be understood as a proof of concept.
>
> That said, scalability depends strongly on the structure of the MEC rather than only on the number of variables. If the MEC contains relatively few admissible pccs, the number of configurations remains manageable even in larger graphs. For example, in our experiments with $d=10$, the maximum number of configurations was 40, so that the order of 100 pccs should still be handled by SkewD.
>
> Finally, a practical way to further improve scalability is to omit the ECM optimization step, which reduces runtime at the cost of some accuracy. In the bivariate setting, this heuristic has shown competitive performance, suggesting it may be a reasonable trade-off in larger problems.

---

> > ### Author Rebuttal · Reviewer_SiZh · 2026-04-03
> >
> > My concerns have been well-addressed. I keep my score unchanged.

---

### Official Review · Reviewer_DEcn · 2026-03-13

**Soundness:** 2
**Presentation:** 3
**Significance:** 3
**Originality:** 3
**Overall Recommendation:** 4
**Confidence:** 5

**Summary:**

This paper addresses the problem of causal discovery in location-scale noise models (LSNMs). This paper argues that existing methods, which typically assume symmetric (often Gaussian) noise distributions, are prone to misspecification when the true noise is skewed. To overcome this limitation, they propose SKEWD, a novel approach that models the noise term using the skew-normal distribution. The paper introduces an estimation procedure for the skew-normal LSNM via an ECM algorithm and presents both bivariate and multivariate variants of SKEWD for causal inference, using either independence tests or likelihood scoring.

**Compliance With Llm Reviewing Policy:**

Affirmed.

**Final Justification:**

Thanks for the response and my concerns have been addressed. I still suggest to futher include the identifiability results to make this paper more complete.

**Key Questions For Authors:**

See the weakness above.

**Limitations:**

yes

**Strengths And Weaknesses:**

Strengths
- The paper investigates an important challenge in current LSNM-based causal discovery methods, which mostly rely on the symmetric noise.
- The demonstration of the paper is clear and well-structured.
- The development of the estimation method for causal discovery under a skew distribution is novel.

Weaknesses
- While the paper mentions that real-world data can be skewed, it does not sufficiently justify why the skew-normal is the most appropriate or sufficient choice. Why focus specifically on this distribution instead of other flexible skew-elliptical families (e.g., skew-t) or non-parametric approaches? A brief discussion on the trade-offs compared to other types of skewed distributions would strengthen the theoretical foundation.
- Although there are some notes on the discussion on the identifiability, this paper lacks a detailed discussion of the identifiability of the proposed skew LSNM. A thorough discussion of the model's identifiability should be included.
- The experimental evaluation would be more robust if it included metrics like F1-score, especially for the multivariate setting. As noted, accuracy can be misleading because it is highly dependent on the sparsity of the graph. Reporting the F1-score would provide a clearer picture of the method's ability.

---

> ### Author Rebuttal · Authors · 2026-03-30
>
> We thank the reviewer for their constructive feedback. We will revise the manuscript to expand the discussion on the choice of the skew-normal distribution (SN) and its implications for identifiability, and include the F1-score (+ precision, recall) in the multivariate experiments.
>
> **Discussion of Identifiability**
>
> Strobl and Lasko (2023) and Immer et al. (2023) have shown that the causal direction in LSNMs is generally identifiable in the bivariate case, except for pathological cases characterized by a specific ordinary differential equation. This includes the SN LSNM. The derived result requires that the conditional log-densities log$(p(y|x))$ in SCM $X\rightarrow Y$ and log$(p(x|y))$ $Y\rightarrow X$ are two-times differentiable, which holds under the SN LSNM. Strobl and Lasko (2023) further extended the theorem to the multivariate case, as discussed in Section 5.3. For clarity we will restate the theorem in the appendix.
>
> **Why Skew-Normal?**
>
> We agree that the choice of the SN deserves further justification. The SN offers several advantages:
>
> - **Natural extension of the Gaussian**: In many fields, errors are assumed to be Gaussian, which is entailed by the SN
> - **Unbounded support**:
> Unlike the exponential, log-normal or gamma distribution, the SN is not restricted to positive values, making it suitable for our application.
> - **Flexible skewness**:
> Compared to other skewed distributions:
> 1. The Gumbel distribution has a fixed skewness (~ 1.14) limiting flexibility, whereas the skew-normal can model skewness within ~ (-1,1).
> 2. The GNO, a generalization of both the normal and log-normal included in our experiments, allows for higher skewness but has truncated tails and is less well-studied.
> 3. The skew-t distribution (Azzalini, 2013) can theoretically model a wider range of skewness, however, its third moment (skewness) only exists for degrees of freedom > 3. It introduces an additional degrees of freedom parameter $\nu$ complicating optimization and unlike the SN, it does not generalize the Gaussian.
> 4. While more general nonparametric or skew-elliptical densities have greater modeling capabilities, they may be too flexible to guarantee accurate estimation and hinder identifiability. We ran experiments using CAREFL (nonparam. via normalizing flows), which performed worse than SkewD, but did not include it as a baseline, since we were unable to reproduce the reported results on the Tübingen dataset, obtaining much lower performance. We will add a discussion on that. Similary, QCCD uses nonparam. quantile regression, but its accuracy heavily degrades with increasing skewness (Fig. 2).
>
> Our experiments show that the SN is perfectly sufficient for the causal discovery task, as SkewD-LL achieves perfect accuracy across all skew-noise datasets, including those generated from the GNO. This includes challenging cases beyond the SN modeling capabilities with skewness 1.75 and excess kurtosis 5.89, comparable to a more heavy-tailed behavior similar to the exponential or t-distr. with $\nu=5$.
>
> Moreover, the SN is well-established in practice across a range of application domains, including environmental science (e.g., modeling air pollution data; [1]), epidemiology (e.g., modeling COVID-19 cases and deaths; [2]), and finance (e.g., asset return modeling; [3]).
>
> Overall, the SN provides a favorable trade-off between flexibility and reliable estimation. It generalizes the Gaussian and is sufficiently expressive to capture relevant asymmetries in practical applications, while remaining simple, smooth, and identifiability-compliant.
>
> **Performance Metrics**
>
> - Bivariate:
> In addition to accuracy, we report the AUDRC, measuring the alignment between confidence and correctness. Since both metrics lead to consistent conclusions, we focused the discussion on accuracy. The F1-score does not provide further information in the bivariate setting, as decisions are forced such that F1-score and accuracy coincide.
> - Multivariate:
> We will include F1-score, precision and recall for all methods. Summary of mean metrics:
> 1. $d=6$
> - SkewD-MV: (F1, Prec., Rec.) = (0.80, 0.84, 0.76)
> - SkewD-MV-O: (0.99, 0.99, 0.99)
> - Best baseline w.r.t. F1-score CAM: (0.28, 0.49, 0.20)
> 2. $d=10$
> - SkewD-MV: (0.74, 0.82, 0.69)
> - SkewD-MV-O: (0.998, 0.998, 0.998)
> - Best baseline w.r.t. F1-score HOST: (0.32, 0.25, 0.46)
> SkewD-MV (KDE) performs slightly worse than SkewD-MV across all settings, which remains the only reliable method.
>
> References:
> [1] Bartoletti, S., & Loperfido, N. (2010). Modelling air pollution data by the skew-normal distribution. Stochastic Environmental Research and Risk Assessment, 24(4), 513-517
> [2] De Leo, S. (2021). Impact of COVID-19 testing strategies and lockdowns on disease management across Europe, South America, and the United States: analysis using skew-Normal distributions. JMIRx Med, 2(2), e21269.
> [3] Carmichael, B., & Coën, A. (2013). Asset pricing with skewed-normal return. Finance Research Letters, 10(2), 50-57.

---

> > ### Author Rebuttal · Reviewer_DEcn · 2026-04-03
> >
> > Thanks for the response and my concerns have been addressed. I still suggest to futher include the identifiability results to make this paper more complete.

---

### Decision · Program_Chairs · 2026-04-30

**Decision:**

Accept (regular)

**Comment:**

This paper addresses causal discovery in Location Scale Noise Models (LSNMs) where the noise distribution may be skewed, a common but previously underexplored scenario. Existing LSNM methods typically assume symmetric (often Gaussian) noise and fail under skewness. The authors propose SKEWD, which models the noise using the skew normal distribution, a natural generalisation that retains the Gaussian as a special case. They develop an estimation procedure via an Expectation Conditional Maximisation (ECM) algorithm combined with a hybrid optimisation strategy (Bayesian optimisation, CMA ES, ECM) to handle the non convex likelihood. For causal inference, SKEWD offers two variants: SKEWD IT (independence testing on residuals) and SKEWD LL (penalised likelihood scoring). The method is extended to multivariate DAG learning via a PC skeleton + scoring approach. Experiments on synthetic and real benchmarks demonstrate that SKEWD achieves near perfect accuracy on highly skewed data while remaining competitive on symmetric data.

All reviewers lean to acceptance (three weak accept, one accept). The paper is technically solid, well motivated, and empirically thorough. Several revisions are required:
* Restate the identifiability theorem statement to the appendix (as promised).

* Include the F1‑score, precision, recall (or other related metrics) for multivariate experiments in the main paper or extended appendix.

* Add a brief discussion of the CAREFL results (even as a footnote) to support the choice of skew‑normal over non‑parametric flows.

* Clarify in the scalability discussion that omitting ECM can be a practical trade‑off for larger graphs.